# When are Kalman-Filter Restless Bandits Indexable?

**Christopher Dance and Tomi Silander**
Xerox Research Centre Europe
6 chemin de Maupertuis, Meylan, Isère, France
{dance,silander}@xrce.xerox.com

## Abstract

We study the restless bandit associated with an extremely simple scalar Kalman filter model in discrete time. Under certain assumptions, we prove that the problem is *indexable* in the sense that the *Whittle index* is a non-decreasing function of the relevant belief state. In spite of the long history of this problem, this appears to be the first such proof. We use results about *Schur-convexity* and *mechanical words*, which are particular binary strings intimately related to *palindromes*.

## 1 Introduction

We study the problem of monitoring several time series so as to maintain a precise belief while minimising the cost of sensing. Such problems can be viewed as POMDPs with belief-dependent rewards [3] and their applications include active sensing [7], attention mechanisms for multiple-object tracking [22], as well as online summarisation of massive data from time-series [4]. Specifically, we discuss the restless bandit [24] associated with the discrete-time Kalman filter [19]. *Restless bandits* generalise bandit problems [6, 8] to situations where the state of each arm (project, site or target) continues to change even if the arm is not played. As with bandit problems, the states of the arms evolve independently given the actions taken, suggesting that there might be efficient algorithms for large-scale settings, based on calculating an *index* for each arm, which is a real number associated with the (belief-)state of that arm alone. However, while bandits always have an optimal index policy (select the arm with the largest index), it is known that no index policy can be optimal for some discrete-state restless bandits [17] and such problems are in general PSPACE-hard even to approximate to any non-trivial factor [10]. Further, in this paper we address restless bandits with real-valued rather than discrete states. On the other hand, Whittle proposed a natural index policy for restless bandits [24], but this policy only makes sense when the restless bandit is *indexable* (Section 2). Briefly, a restless bandit is said to be indexable when an optimal solution to a relaxed version of the problem consists in playing all arms whose indices exceed a given threshold. (The relaxed version of the problem relaxes the constraint on the number of arms pulled per turn to a constraint on the average number of arms pulled per turn). Under certain conditions, indexability implies a form of asymptotic optimality of Whittle's policy for the original problem [23, 20].

Restless bandits associated with scalar Kalman(-Bucy) filters in continuous time were recently shown to be indexable [12] and the corresponding discrete-time problem has attracted considerable attention over a long period [15, 11, 16, 21]. However, that attention has produced no satisfactory proof of indexability – *even* for scalar time-series and even if we assume that there is a *monotone* optimal policy for the single-arm problem, which is a policy that plays the arm if and only if the relevant belief-state exceeds some threshold (here the relevant belief-state is a posterior variance). Theorem 1 of this paper addresses that gap. After formalising the problem (Section 2), we describe the concepts and intuition (Section 3) behind the main result (Section 4). The main tools are *mechanical words* (which are not sufficiently well-known) and *Schur convexity*. As these tools are associated with rather general theorems, we believe that future work (Section 5) should enable substantial generalisation of our results.

## 2 Problem and Index

We consider the problem of tracking $N$ time-series, which we call arms, in discrete time. The state $Z_{i,t} \in \mathbb{R}$ of arm $i$ at time $t \in \mathbb{Z}_+$ evolves as a standard-normal random walk independent of everything but its immediate past ($\mathbb{Z}_+, \mathbb{R}_-$ and $\mathbb{R}_+$ all include zero). The action space is $\mathcal{U} := \{1, \dots, N\}$. Action $u_t = i$ makes an expensive observation $Y_{i,t}$ of arm $i$ which is normally-distributed about $Z_{i,t}$ with precision $b_i \in \mathbb{R}_+$ and we receive cheap observations $Y_{j,t}$ of each other arm $j$ with precision $a_j \in \mathbb{R}_+$ where $a_j < b_j$ and $a_j = 0$ means no observation at all.

Let $Z_t, Y_t, \mathcal{H}_t, \mathcal{F}_t$ be the state, observation, history and observed history, so that $Z_t := (Z_{1,t}, \dots, Z_{N,t}), Y_t := (Y_{1,t}, \dots, Y_{N,t}), \mathcal{H}_t := ((Z_0, u_0, Y_0), \dots, (Z_t, u_t, Y_t))$ and $\mathcal{F}_t := ((u_0, Y_0), \dots, (u_t, Y_t))$. Then we formalise the above as ($\mathbf{1}$. is the indicator function)

$$Z_{i,0} \sim \mathcal{N}(0,1), \quad Z_{i,t+1} \mid \mathcal{H}_t \sim \mathcal{N}(Z_{i,t}, 1), \quad Y_{i,t} \mid \mathcal{H}_{t-1}, Z_t, u_t \sim \mathcal{N}\left(Z_{i,t}, \frac{\mathbf{1}_{u_t \neq i}}{a_i} + \frac{\mathbf{1}_{u_t = i}}{b_i}\right).$$

Note that this setting is readily generalised to $\mathbb{E}[(Z_{i,t+1} - Z_{i,t})^2] \neq 1$ by a change of variables.

Thus the posterior belief is given by the Kalman filter as $Z_{i,t} \mid \mathcal{F}_t \sim \mathcal{N}(\hat{Z}_{i,t}, x_{i,t})$ where the posterior mean is $\hat{Z}_{i,t} \in \mathbb{R}$ and the *error variance* $x_{i,t} \in \mathbb{R}_+$ satisfies

$$x_{i,t+1} = \phi_{i,\mathbf{1}_{u_{t+1}=i}}(x_{i,t}) \quad \text{where} \quad \phi_{i,0}(x) := \frac{x+1}{a_i x + a_i + 1} \text{ and } \phi_{i,1}(x) := \frac{x+1}{b_i x + b_i + 1}. \quad (1)$$

**Problem KF1.** Let $\pi$ be a policy so that $u_t = \pi(\mathcal{F}_{t-1})$. Let $x_{i,t}^{\pi}$ be the error variance under $\pi$. The problem is to choose $\pi$ so as to minimise the following objective for discount factor $\beta \in [0,1)$. The objective consists of a weighted sum of error variances $x_{i,t}^{\pi}$ with weights $w_i \in \mathbb{R}_+$ plus observation costs $h_i \in \mathbb{R}_+$ for $i = 1, \dots, N$:

$$\mathbb{E}\left[\sum_{t=0}^{\infty} \sum_{i=1}^{N} \beta^t \left\{h_i \mathbf{1}_{u_t=i} + w_i x_{i,t}^{\pi}\right\}\right] = \sum_{t=0}^{\infty} \sum_{i=1}^{N} \beta^t \left\{h_i \mathbf{1}_{u_t=i} + w_i x_{i,t}^{\pi}\right\}$$

where the equality follows as (1) is a deterministic mapping (and assuming $\pi$ is deterministic).

**Single-Arm Problem and Whittle Index.** Now fix an arm $i$ and write $x_t^{\pi}, \phi_0(\cdot), \dots$ instead of $x_{t,i}^{\pi}, \phi_{i,0}(\cdot), \dots$. Say there are now two actions $u_t = 0, 1$ corresponding to cheap and expensive observations respectively and the expensive observation now costs $h + \nu$ where $\nu \in \mathbb{R}$. The *single-arm problem* is to choose a policy, which here is an action sequence, $\pi := (u_0, u_1, \dots)$

$$\text{so as to minimise} \quad V^{\pi}(x|\nu) := \sum_{t=0}^{\infty} \beta^t \left\{(h+\nu)u_t + wx_t^{\pi}\right\} \quad \text{where } x_0 = x. \quad (2)$$

Let $Q(x, \alpha|\nu)$ be the optimal cost-to-go in this problem if the first action must be $\alpha$ and let $\pi^*$ be an optimal policy, so that

$$Q(x, \alpha|\nu) := (h+\nu)\alpha + wx + \beta V^{\pi^*}(\phi_\alpha(x)|\nu).$$

For any fixed $x \in \mathbb{R}_+$, the value of $\nu$ for which actions $u_0 = 0$ and $u_0 = 1$ are both optimal is known as the *Whittle index* $\lambda^W(x)$ assuming it exists and is unique. In other words

*The Whittle index $\lambda^W(x)$ is the solution to $Q(x, 0|\lambda^W(x)) = Q(x, 1|\lambda^W(x))$.* $\quad (3)$

Let us consider a policy which takes action $u_0 = \alpha$ then acts optimally producing actions $u_t^{\alpha*}(x)$ and error variances $x_t^{\alpha*}(x)$. Then (3) gives

$$\sum_{t=0}^{\infty} \beta^t \left\{(h + \lambda^W(x))u_t^{0*} + wx_t^{0*}(x)\right\} = \sum_{t=0}^{\infty} \beta^t \left\{(h + \lambda^W(x))u_t^{1*} + wx_t^{1*}(x)\right\}.$$

Solving this linear equation for the index $\lambda^W(x)$ gives

$$\lambda^W(x) = w \frac{\sum_{t=1}^{\infty} \beta^t (x_t^{0*}(x) - x_t^{1*}(x))}{\sum_{t=0}^{\infty} \beta^t (u_t^{1*}(x) - u_t^{0*}(x))} - h. \quad (4)$$

Whittle [24] recognised that for his index policy (play the arm with the largest $\lambda^W(x)$) to make sense, any arm which receives an expensive observation for added cost $\nu$, must also receive an expensive observation for added cost $\nu' < \nu$. Such problems are said to be *indexable*. The question resolved by this paper is whether Problem KF1 is indexable. Equivalently, is $\lambda^W(x)$ non-decreasing in $x \in \mathbb{R}_+$?

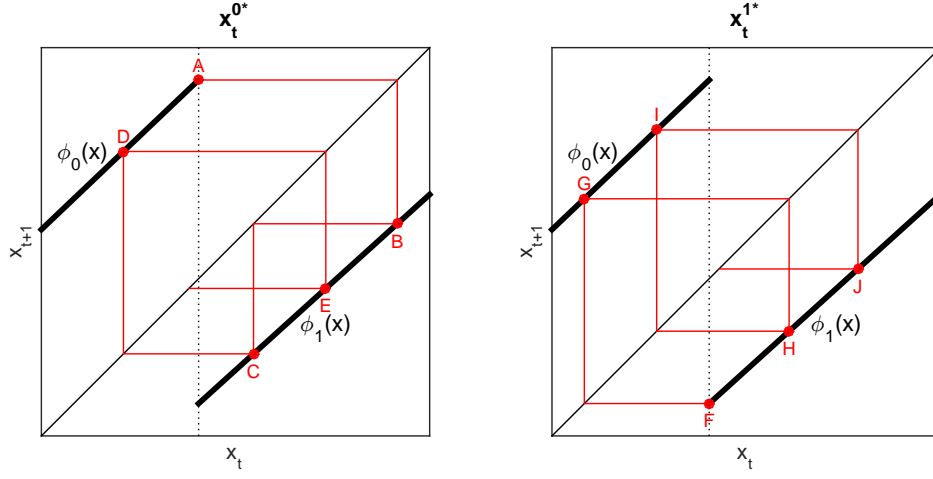

Figure 1: Orbit $x_t^{0*}(x)$ traces the path $ABCDE\ldots$ for the word $01w = 01101$. Orbit $x_t^{1*}(x)$ traces the path $FGHIJ\ldots$ for the word $10w = 10101$. Word $w = 101$ is a palindrome.

## 3 Main Result, Key Concepts and Intuition

We make the following intuitive assumption about threshold (monotone) policies.

**A1.** *For some $x \in \mathbb{R}_+$ depending on $\nu \in \mathbb{R}$, the policy $u_t = \mathbf{1}_{x_t \geq x}$ is optimal for problem (2).*

Note that under A1, definition (3) means the policy $u_t = \mathbf{1}_{x_t > x}$ is also optimal, so we can choose

$$\left.\begin{aligned}
u_t^{0*}(x) &:= \begin{cases} 0 & \text{if } x_{t-1}^{0*}(x) \leq x \\ 1 & \text{otherwise} \end{cases} \quad \text{and} \quad x_t^{0*}(x) := \begin{cases} \phi_0(x_{t-1}^{0*}(x)) & \text{if } x_{t-1}^{0*}(x) \leq x \\ \phi_1(x_{t-1}^{0*}(x)) & \text{otherwise} \end{cases} \\
u_t^{1*}(x) &:= \begin{cases} 0 & \text{if } x_{t-1}^{1*}(x) < x \\ 1 & \text{otherwise} \end{cases} \quad \text{and} \quad x_t^{1*}(x) := \begin{cases} \phi_0(x_{t-1}^{1*}(x)) & \text{if } x_{t-1}^{1*}(x) < x \\ \phi_1(x_{t-1}^{1*}(x)) & \text{otherwise} \end{cases}
\end{aligned}\right\} \quad (5)$$

where $x_0^{0*}(x) = x_0^{1*}(x) = x$. We refer to $x_t^{0*}(x), x_t^{1*}(x)$ as the *x-threshold orbits* (Figure 1).

We are now ready to state our main result.

**Theorem 1.** *Suppose a threshold policy (A1) is optimal for the single-arm problem (2). Then Problem KF1 is indexable. Specifically, for any $b > a \geq 0$ let*

$$\phi_0(x) := \frac{x+1}{ax+a+1}, \qquad\qquad \phi_1(x) := \frac{x+1}{bx+b+1}$$

*and for any $w \in \mathbb{R}_+, h \in \mathbb{R}$ and $0 < \beta < 1$, let*

$$\lambda^W(x) := w \frac{\sum_{t=1}^{\infty} \beta^t (x_t^{0*}(x) - x_t^{1*}(x))}{\sum_{t=0}^{\infty} \beta^t (u_t^{1*}(x) - u_t^{0*}(x))} - h \qquad (6)$$

*in which action sequences $u_t^{0*}(x), u_t^{1*}(x)$ and error variance sequences $x_t^{0*}(x), x_t^{1*}(x)$ are given in terms of $\phi_0, \phi_1$ by (5). Then $\lambda^W(x)$ is a continuous and non-decreasing function of $x \in \mathbb{R}_+$.*

We are now ready to describe the key concepts underlying this result.

**Words.** In this paper, a *word* $w$ is a string on $\{0,1\}^*$ with $k^{\text{th}}$ letter $w_k$ and $w_{i:j} := w_i w_{i+1} \ldots w_j$. The empty word is $\epsilon$, the concatenation of words $u, v$ is $uv$, the word that is the $n$-fold repetition of $w$ is $w^n$, the infinite repetition of $w$ is $w^\omega$ and $\tilde{w}$ is the reverse of $w$, so $w = \tilde{w}$ means $w$ is a palindrome. The length of $w$ is $|w|$ and $|w|_u$ is the number of times that word $u$ appears in $w$, overlaps included.

**Christoffel, Sturmian and Mechanical Words.** It turns out that the action sequences in (5) are given by such words, so the following definitions are central to this paper.

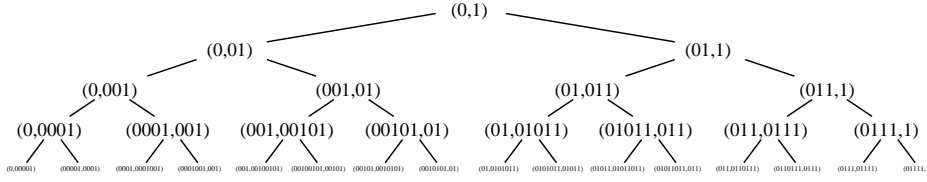

Figure 2: Part of the Christoffel tree.

The *Christoffel tree* (Figure 2) is an infinite complete binary tree [5] in which each node is labelled with a pair $(u, v)$ of words. The root is $(0, 1)$ and the children of $(u, v)$ are $(u, uv)$ and $(uv, v)$. The *Christoffel words* are the words $0, 1$ and the concatenations $uv$ for all $(u, v)$ in that tree. The fractions $|uv|_1/|uv|_0$ form the Stern-Brocot tree [9] which contains each positive rational number exactly once. Also, infinite paths in the Stern-Brocot tree converge to the positive irrational numbers. Analogously, *Sturmian words* could be thought of as infinitely-long Christoffel words.

Alternatively, among many known characterisations, the Christoffel words can be defined as the words $0, 1$ and the words $0w1$ where $a := |0w1|_1/|0w1|$ and

$$(01w)_n := \lfloor (n+1)a \rfloor - \lfloor na \rfloor$$

for any relatively prime natural numbers $|0w1|_0$ and $|0w1|_1$ and for $n = 1, 2, \ldots, |0w1|$. The Sturmian words are then the infinite words $0w_1 w_2 \cdots$ where, for $n = 1, 2, \ldots$ and $a \in (0, 1)\backslash \mathbb{Q}$,

$$(01w_1 w_2 \cdots)_n := \lfloor (n+1)a \rfloor - \lfloor na \rfloor.$$

We use the notation $0w1$ for Sturmian words although they are infinite.

The set of *mechanical words* is the union of the Christoffel and Sturmian words [13]. (Note that the mechanical words are sometimes defined in terms of infinite repetitions of the Christoffel words.)

**Majorisation.** As in [14], let $x, y \in \mathbb{R}^m$ and let $x_{(i)}$ and $y_{(i)}$ be their elements sorted in *ascending* order. We say $x$ is *weakly supermajorised* by $y$ and write $x \prec^w y$ if

$$\sum_{k=1}^{j} x_{(k)} \geq \sum_{k=1}^{j} y_{(k)} \qquad \text{for all } j = 1, \ldots, m.$$

If this is an equality for $j = m$ we say $x$ is *majorised* by $y$ and write $x \prec y$. It turns out that

$$x \prec y \qquad \Leftrightarrow \qquad \sum_{k=1}^{j} x_{[k]} \leq \sum_{k=1}^{j} y_{[k]} \quad \text{for } j = 1, \ldots, m-1 \text{ with equality for } j = m$$

where $x_{[k]}, y_{[k]}$ are the sequences sorted in *descending* order. For $x, y \in \mathbb{R}^m$ we have [14]

$$x \prec y \qquad \Leftrightarrow \qquad \sum_{i=1}^{m} f(x_i) \leq \sum_{i=1}^{m} f(y_i) \quad \text{for all convex functions } f : \mathbb{R} \to \mathbb{R}.$$

More generally, a real-valued function $\phi$ defined on a subset $\mathcal{A}$ of $\mathbb{R}^m$ is said to be *Schur-convex* on $\mathcal{A}$ if $x \prec y$ implies that $\phi(x) \leq \phi(y)$.

**Möbius Transformations.** Let $\mu_A(x)$ denote the Möbius transformation $\mu_A(x) := \frac{A_{11}x + A_{12}}{A_{21}x + A_{22}}$ where $A \in \mathbb{R}^{2 \times 2}$. Möbius transformations such as $\phi_0(\cdot), \phi_1(\cdot)$ are closed under composition, so for any word $w$ we define $\phi_w(x) := \phi_{w_{|w|}} \circ \cdots \circ \phi_{w_2} \circ \phi_{w_1}(x)$ and $\phi_\epsilon(x) := x$.

**Intuition.** Here is the intuition behind our main result.

For any $x \in \mathbb{R}_+$, the orbits in (5) correspond to a particular mechanical word $0, 1$ or $0w1$ depending on the value of $x$ (Figure 1). Specifically, for any word $u$, let $y_u$ be the fixed point of the mapping $\phi_u$ on $\mathbb{R}_+$ so that $\phi_u(y_u) = y_u$ and $y_u \in \mathbb{R}_+$. Then the word corresponding to $x$ is $1$ for $0 \leq x \leq y_1$, $0w1$ for $x \in [y_{01w}, y_{10w}]$ and $0$ for $y_0 \leq x < \infty$. In passing we note that these fixed points are sorted in ascending order by the ratio $\rho := |01w|_0/|01w|_1$ of counts of 0s to counts of 1s, as

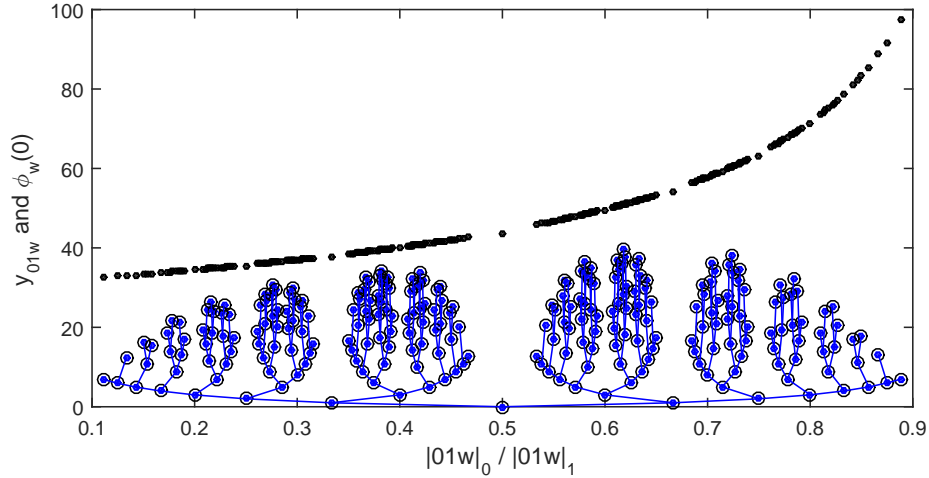

Figure 3: Lower fixed points $y_{01w}$ of Christoffel words (black dots), majorisation points for those words (black circles) and the tree of $\phi_w(0)$ (blue).

illustrated by Figure 3. Interestingly, it turns out that ratio $\rho$ is a piecewise-constant yet continuous function of $x$, reminiscent of the Cantor function.

Also, composition of Möbius transformations is homeomorphic to matrix multiplication so that

$$\mu_A \circ \mu_B(x) = \mu_{AB}(x) \qquad \text{for any } A, B \in \mathbb{R}^{2 \times 2}.$$

Thus, the index (6) can be written in terms of the orbits of a linear system (11) given by $0, 1$ or $0w1$. Further, if $A \in \mathbb{R}^{2 \times 2}$ and $\det(A) = 1$ then the gradient of the corresponding Möbius transformation is the convex function

$$\frac{d\mu_A(x)}{dx} = \frac{1}{(A_{21}x + A_{22})^2}.$$

So the gradient of the index is the difference of the sums of a convex function of the linear-system orbits. However, such sums are Schur-convex functions and it follows that the index is increasing because one orbit weakly supermajorises the other, as we now show for the case $0w1$ (noting that the proof is easier for words $0, 1$). As $0w1$ is a mechanical word, $w$ is a palindrome. Further, if $w$ is a palindrome, it turns out that the difference between the linear-system orbits increases with $x$. So, we might define the *majorisation point* for $w$ as the $x$ for which one orbit majorises the other. Quite remarkably, if $w$ is a palindrome then the majorisation point is $\phi_w(0)$ (Proposition 7). Indeed the black circles and blue dots of Figure 3 coincide. Finally, $\phi_w(0)$ is less than or equal to $y_{01w}$ which is the least $x$ for which the orbits correspond to the word $0w1$. Indeed, the blue dots of Figure 3 are below the corresponding black dots. Thus one orbit does indeed supermajorise the other.

## 4 Proof of Main Result

### 4.1 Mechanical Words

The Möbius transformations of (1) satisfy the following assumption for $\mathcal{I} := \mathbb{R}_+$. We prove that the fixed point $y_w$ of word $w$ (the solution to $\phi_w(x) = x$ on $\mathcal{I}$) is unique in the supplementary material.

**Assumption A2.** *Functions $\phi_0 : \mathcal{I} \to \mathcal{I}, \phi_1 : \mathcal{I} \to \mathcal{I}$, where $\mathcal{I}$ is an interval of $\mathbb{R}$, are increasing and non-expansive, so for all $x, y \in \mathcal{I} : x < y$ and for $k \in \{0, 1\}$ we have*

$$\underbrace{\phi_k(x) < \phi_k(y)}_{increasing} \qquad and \qquad \underbrace{\phi_k(y) - \phi_k(x) < y - x}_{non\text{-}expansive}.$$

*Furthermore, the fixed points $y_0, y_1$ of $\phi_0, \phi_1$ on $\mathcal{I}$ satisfy $y_1 < y_0$.*

Hence the following two propositions (supplementary material) apply to $\phi_0, \phi_1$ of (1) on $\mathcal{I} = \mathbb{R}_+$.

**Proposition 1.** *Suppose A2 holds, $x \in \mathcal{I}$ and $w$ is a non-empty word. Then*

$$x < \phi_w(x) \Leftrightarrow \phi_w(x) < y_w \Leftrightarrow x < y_w \quad \text{and} \quad x > \phi_w(x) \Leftrightarrow \phi_w(x) > y_w \Leftrightarrow x > y_w.$$

For a given $x$, in the notation of (5), we call the shortest word $u$ such that $(u_1^{1*}, u_2^{1*}, \dots) = u^\omega$ the *$x$-threshold word*. Proposition 2 generalises a recent result about $x$-threshold words in a setting where $\phi_0, \phi_1$ are linear [18].

**Proposition 2.** *Suppose A2 holds and $0w1$ is a mechanical word. Then*

$$0w1 \text{ is the } x\text{-threshold word} \Leftrightarrow x \in [y_{01w}, y_{10w}].$$

*Also, if $x_0, x_1 \in \mathcal{I}$ with $x_0 \geq y_0$ and $x_1 \leq y_1$ then the $x_0$- and $x_1$-threshold words are $0$ and $1$.*

We also use the following very interesting fact (Proposition 4.2 on p.28 of [5]).

**Proposition 3.** *Suppose $0w1$ is a mechanical word. Then $w$ is a palindrome.*

## 4.2 Properties of the Linear-System Orbits $M(w)$ and Prefix Sums $S(w)$

**Definition.** Assume that $a, b \in \mathbb{R}_+$ and $a < b$. Consider the matrices

$$F := \begin{pmatrix} 1 & 1 \\ a & 1+a \end{pmatrix}, \qquad G := \begin{pmatrix} 1 & 1 \\ b & 1+b \end{pmatrix} \quad \text{and} \quad K := \begin{pmatrix} -1 & -1 \\ 0 & 1 \end{pmatrix}$$

so that the Möbius transformations $\mu_F, \mu_G$ are the functions $\phi_0, \phi_1$ of (1) and $GF - FG = (b-a)K$. Given any word $w \in \{0,1\}^*$, we define the *matrix product $M(w)$*

$$M(w) := M(w_{|w|}) \cdots M(w_1), \quad \text{where } M(\epsilon) := I, M(0) := F \text{ and } M(1) := G$$

where $I \in \mathbb{R}^{2 \times 2}$ is the identity and the *prefix sum $S(w)$* as the matrix polynomial

$$S(w) := \sum_{k=1}^{|w|} M(w_{1:k}), \qquad \text{where } S(\epsilon) := 0 \text{ (the all-zero matrix).} \tag{7}$$

For any $A \in \mathbb{R}^{2 \times 2}$, let $\mathrm{tr}(A)$ be the trace of $A$, let $A_{ij} = [A]_{ij}$ be the entries of $A$ and let $A \geq 0$ indicate that all entries of $A$ are non-negative.

**Remark.** Clearly, $\det(F) = \det(G) = 1$ so that $\det(M(w)) = 1$ for any word $w$. Also, $S(w)$ corresponds to the partial sums of the linear-system orbits, as hinted in the previous section.

The following proposition captures the role of palindromes (proof in the supplementary material).

**Proposition 4.** *Suppose $w$ is a word, $p$ is a palindrome and $n \in \mathbb{Z}_+$. Then*

1. *$M(p) = \begin{pmatrix} \frac{fh+1}{h+f} & f \\ \frac{h^2-1}{h+f} & h \end{pmatrix}$ for some $f, h \in \mathbb{R}$,*

2. *$\mathrm{tr}(M(10p)) = \mathrm{tr}(M(01p))$,*

3. *If $u \in \{p(10p)^n, (10p)^n 10\}$ then $M(u) - M(\tilde{u}) = \lambda K$ for some $\lambda \in \mathbb{R}_-$,*

4. *If $w$ is a prefix of $p$ then $[M(p(10p)^n 10w)]_{22} \leq [M(p(01p)^n 01w)]_{22}$,*

5. *$[M((10p)^n 10w)]_{21} \geq [M((01p)^n 01w)]_{21}$,*

6. *$[M((10p)^n 1)]_{21} \geq [M((01p)^n 0)]_{21}$.*

We now demonstrate a surprisingly simple relation between $S(w)$ and $M(w)$.

**Proposition 5.** *Suppose $w$ is a palindrome. Then*

$$S_{21}(w) = M_{22}(w) - 1 \quad \text{and} \quad S_{22}(w) = M_{12}(w) + S_{21}(w). \tag{8}$$

*Furthermore, if $\Delta_k := [S(10w)M(w(10w)^k) - S(01w)M(w(01w)^k)]_{22}$ then*

$$\Delta_k = 0 \quad \text{for all } k \in \mathbb{Z}_+. \tag{9}$$

*Proof.* Let us write $M := M(w), S := S(w)$. We prove (8) by induction on $|w|$. In the base case $w \in \{\epsilon, 0, 1\}$. For $w = \epsilon$, $M_{22} - 1 = 0 = S_{21}, M_{12} + S_{21} = 0 = S_{22}$. For $w \in \{0, 1\}$, $M_{22} - 1 = c = S_{21}, M_{12} + S_{21} = 1 + c = S_{22}$ for some $c \in \{a, b\}$. For the inductive step, in accordance with Claim 1 of Proposition 4, assume $w \in \{0v0, 1v1\}$ for some word $v$ satisfying

$$M(v) = \begin{pmatrix} \frac{fh+1}{h+f} & f \\ \frac{h^2-1}{h+f} & h \end{pmatrix}, \qquad S(v) = \begin{pmatrix} c & d \\ h-1 & f+h-1 \end{pmatrix} \quad \text{for some } c, d, f, h \in \mathbb{R}.$$

For $w = 1v1$, $M := M(1v1) = GM(v)G$ and $S := S(1v1) = GM(v)G + S(v)G + G$. Calculating the corresponding matrix products and sums gives

$$S_{21} = (bh + h + bf - 1)(bh + 2h + bf + f + 1)(h + f)^{-1} = M_{22} - 1$$
$$S_{22} - S_{21} = bh + 2h + bf + f = M_{12}$$

as claimed. For $w = 0u0$ the claim also holds as $F = G|_{b=a}$. This completes the proof of (8).

*Furthermore Part.* Let $A := S(w)FG + FG + G$ and $B := S(w)GF + GF + F$. Then

$$\Delta_k = [(A(M(w)FG)^k - B(M(w)GF)^k)M(w)]_{22} \tag{10}$$

by definition of $S(\cdot)$. By Claim 1 of Proposition 4 and (8) we know that

$$M(w) = \begin{pmatrix} \frac{fh+1}{h+f} & f \\ \frac{h^2-1}{h+f} & h \end{pmatrix}, \qquad S(w) = \begin{pmatrix} c & d \\ h-1 & f+h-1 \end{pmatrix} \quad \text{for some } c, d, f, h \in \mathbb{R}.$$

Substituting these expressions and the definitions of $F, G$ into the definitions of $A, B$ and then into (10) for $k \in \{0, 1\}$ directly gives $\Delta_0 = \Delta_1 = 0$ (although this calculation is long).

Now consider the case $k \geq 2$. Claim 2 of Proposition 4 says $\text{tr}(M(10w)) = \text{tr}(M(01w))$ and clearly $\det(M(10w)) = \det(M(01w)) = 1$. Thus we can diagonalise as

$$M(w)FG =: UDU^{-1}, \quad M(w)GF =: VDV^{-1}, \quad D := \text{diag}(\lambda, 1/\lambda) \quad \text{for some } \lambda \geq 1$$

so that $\Delta_k = [AUD^kU^{-1}M(w) - e^T BVD^kV^{-1}M(w)]_{22} =: \gamma_1\lambda^k + \gamma_2\lambda^{-k}$. So, if $\lambda = 1$ then $\Delta_k = \gamma_1 + \gamma_2 = \Delta_0$ and we already showed that $\Delta_0 = 0$. Otherwise $\lambda \neq 1$, so $\Delta_0 = \Delta_1 = 0$ implies $\gamma_1 + \gamma_2 = \gamma_1\lambda + \gamma_2\lambda^{-1} = 0$ which gives $\gamma_1 = \gamma_2 = 0$. Thus for any $k \in \mathbb{Z}_+$ we have $\Delta_k = \gamma_1\lambda^k + \gamma_2\lambda^{-k} = 0$. $\square$

### 4.3 Majorisation

The following is a straightforward consequence of results in [14] proved in the supplementary material. We emphasize that the notation $\prec^w$ has nothing to do with the notion of $w$ as a word.

**Proposition 6.** *Suppose $x, y \in \mathbb{R}_+^m$ and $f : \mathbb{R} \to \mathbb{R}$ is a symmetric function that is convex and decreasing on $\mathbb{R}_+$. Then $x \prec^w y$ and $\beta \in [0, 1] \quad \Rightarrow \quad \sum_{i=1}^m \beta^i f(x_{(i)}) \geq \sum_{i=1}^m \beta^i f(y_{(i)})$.*

For any $x \in \mathbb{R}$ and any fixed word $w$, define the sequences for $n \in \mathbb{Z}_+$ and $k = 1, \ldots, m$

$$\left. \begin{array}{l} x_{nm+k}(x) := [M((10w)^n(10w)_{1:k})v(x)]_2, \quad \sigma_x^{(n)} := (x_{nm+1}(x), \ldots, x_{nm+m}(x)) \\ y_{nm+k}(x) := [M((01w)^n(01w)_{1:k})v(x)]_2, \quad \sigma_y^{(n)} := (y_{nm+1}(x), \ldots, y_{nm+m}(x)) \end{array} \right\} \tag{11}$$

where $m := |10w|$ and $v(x) := (x, 1)^T$.

**Proposition 7.** *Suppose $w$ is a palindrome and $x \geq \phi_w(0)$. Then $\sigma_x^{(n)}$ and $\sigma_y^{(n)}$ are ascending sequences on $\mathbb{R}_+$ and $\sigma_x^{(n)} \prec^w \sigma_y^{(n)}$ for any $n \in \mathbb{Z}_+$.*

*Proof.* Clearly $\phi_w(0) \geq 0$ so $x \geq 0$ and hence $v(x) \geq 0$. So for any word $u$ and letter $c \in \{0, 1\}$ we have $M(uc)v(x) = M(c)M(u)v(x) \geq M(u)v(x) \geq 0$ as $M(c) \geq I$. Thus $x_{k+1}(x) \geq x_k(x) \geq 0$ and $y_{k+1}(x) \geq y_k(x) \geq 0$. In conclusion, $\sigma_x^{(n)}$ and $\sigma_y^{(n)}$ are ascending sequences on $\mathbb{R}_+$.

Now $\phi_w(0) = \frac{[M(w)]_{12}}{[M(w)]_{22}}$. Thus $[Av(\phi_w(0))]_2 := \frac{[AM(w)]_{22}}{[M(w)]_{22}}$ for any $A \in \mathbb{R}^{2\times 2}$. So

$$x_{nm+k}(\phi_w(0)) - y_{nm+k}(\phi_w(0))$$
$$= \frac{1}{[M(w)]_{22}} [(M((10w)^n(10w)_{1:k}) - M((01w)^n(01w)_{1:k}))M(w)]_{22} \leq 0$$

for $k = 2, \ldots, m$ by Claim 4 of Proposition 4. So all but the first term of the sum $T_m(\phi_w(0))$ is non-positive where

$$T_j(x) := \sum_{k=1}^{j} (x_{nm+k}(x) - y_{nm+k}(x)).$$

Thus $T_1(\phi_w(0)) \geq T_2(\phi_w(0)) \geq \ldots T_m(\phi_w(0))$. But

$$T_m(\phi_w(0)) = \frac{1}{[M(w)]_{22}} \sum_{k=1}^{m} [(M((10w)^n (10w)_{1:k}) - M((01w)^n (01w)_{1:k})) M(w)]_{22}$$

$$= \frac{1}{[M(w)]_{22}} [S(10w) M(w(10w)^n) - S(01w) M(w(01w)^n)]_{22} = 0$$

where the last step follows from (9). So $T_j(\phi_w(0)) \geq 0$ for $j = 1, \ldots, m$. Yet Claims 5 and 6 of Proposition 4 give $\frac{d}{dx} T_j(x) = \sum_{k=1}^{j} [M((10w)^n (10w)_{1:k}) - M((01w)^n (01w)_{1:k})]_{21} \geq 0$. So for $x \geq \phi_w(0)$ we have $T_j(x) \geq 0$ for $j = 1, \ldots, m$ which means that $\sigma_x^{(n)} \prec^w \sigma_y^{(n)}$. $\square$

### 4.4 Indexability

**Theorem 1.** *The index $\lambda^W(x)$ of (6) is continuous and non-decreasing for $x \in \mathbb{R}_+$.*

*Proof.* As weight $w$ is non-negative and cost $h$ is a constant we only need to prove the result for $\lambda(x) := \lambda^W(x)\big|_{w=1,h=0}$ and we can use $w$ to denote a word. By Proposition 2, $x \in [y_{01w}, y_{10w}]$ for some mechanical word $0w1$. (Cases $x \notin (y_1, y_0)$ are clarified in the supplementary material.)

Let us show that the hypotheses of Proposition 7 are satisfied by $w$ and $x$. Firstly, $w$ is a palindrome by Proposition 3. Secondly, $\phi_{w01}(0) \geq 0$ and as $\phi_w(\cdot)$ is monotonically increasing, it follows that $\phi_w \circ \phi_{w01}(0) \geq \phi_w(0)$. Equivalently, $\phi_{01w} \circ \phi_w(0) \geq \phi_w(0)$ so that $\phi_w(0) \leq y_{01w}$ by Proposition 1. Hence $x \geq y_{01w} \geq \phi_w(0)$.

Thus Proposition 7 applies, showing that the sequences $\sigma_x^{(n)}$ and $\sigma_y^{(n)}$, with elements $x_{nm+k}(x)$ and $y_{nm+k}(x)$ as defined in (11), are non-decreasing sequences on $\mathbb{R}_+$ with $\sigma_x^{(n)} \prec^w \sigma_y^{(n)}$. Also, $1/x^2$ is a symmetric function that is convex and decreasing on $\mathbb{R}_+$. Therefore Proposition 6 applies giving

$$\sum_{k=1}^{m} \left( \frac{\beta^{nm+k-1}}{(x_{nm+k}(x))^2} - \frac{\beta^{nm+k-1}}{(y_{nm+k}(x))^2} \right) \geq 0 \qquad \text{for any } n \in \mathbb{Z}_+ \text{ where } m := |01w|. \tag{12}$$

Also Proposition 2 shows that the $x$-threshold orbits are $(\phi_{u_1}(x), \ldots, \phi_{u_{1:k}}(x), \ldots)$ and $(\phi_{l_1}(x), \ldots, \phi_{l_{1:k}}(x), \ldots)$ where $u := (01w)^\omega$ and $l := (10w)^\omega$. So the denominator of (6) is

$$\sum_{k=0}^{\infty} \beta^k (\mathbb{1}_{l_{k+1}=1} - \mathbb{1}_{u_{k+1}=1}) = \sum_{k=0}^{\infty} \beta^{mk}(1-\beta) \Rightarrow \lambda(x) = \frac{1-\beta^m}{1-\beta} \sum_{k=1}^{\infty} \beta^{k-1} (\phi_{u_{1:k}}(x) - \phi_{l_{1:k}}(x)).$$

Note that $\frac{d}{dx} \frac{ex+f}{gx+h} = \frac{1}{(gx+h)^2}$ for any $eh - fg = 1$. Then (12) gives

$$\frac{d\lambda(x)}{dx} = \frac{1-\beta^m}{1-\beta} \sum_{n=0}^{\infty} \sum_{k=1}^{m} \left( \frac{\beta^{nm+k-1}}{(x_{nm+k}(x))^2} - \frac{\beta^{nm+k-1}}{(y_{nm+k}(x))^2} \right) \geq 0.$$

But $\lambda(x)$ is continuous for $x \in \mathbb{R}_+$ (as shown in the supplementary material). Therefore we conclude that $\lambda(x)$ is non-decreasing for $x \in \mathbb{R}_+$. $\square$

## 5   Further Work

One might attempt to prove that assumption A1 holds using general results about monotone optimal policies for two-action MDPs based on submodularity [2] or multimodularity [1]. However, we find counter-examples to the required submodularity condition. Rather, we are optimistic that the ideas of this paper themselves offer an alternative approach to proving A1. It would then be natural to extend our results to settings where the underlying state evolves as $Z_{t+1} \mid \mathcal{H}_t \sim \mathcal{N}(mZ_t, 1)$ for some multiplier $m \neq 1$ and to cost functions other than the variance. Finally, the question of the indexability of the discrete-time Kalman filter in multiple dimensions remains open.

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
