[Supplementary Material]

# Supplementary Material

**Christopher Dance and Tomi Silander**
Xerox Research Centre Europe
6 chemin de Maupertuis, Meylan, Isère, France
{dance,silander}@xrce.xerox.com

## 1 Introduction

The results used but not proved in the main paper are given here as:

- Proposition 2 which was used to show that $\phi_w(0) \leq x$,
- Proposition 9 for the range of $x$ giving a specific mechanical word,
- Proposition 10 showing the index is continuous for $x \in \mathbb{R}_+$,
- Proposition 12 showing the properties of $M(p)$ when $p$ is a palindrome.
- and Proposition 13 for weak supermajorisation with $\beta \neq 1$.

A clarification of the extreme cases of Theorem 1 of the main paper and an analysis of the computational cost of approximating the index are presented in the final sections.

## 2 From $x$-Threshold Policies to Mechanical Words

Some concepts relating to mechanical words appeared as early as 1771 in Jean Bernoulli's study of continued fractions (Berstel *et al*, 2008). The term "mechanical sequences" appears in the work of Morse and Hedlund (Am. J. Math., Vol 62, No. 1, 1940, p. 1-42) who had just introduced the term "symbolic dynamics". Morse and Hedlund studied the concept from the perspective of sequences of the form $\lfloor c + k\beta \rfloor$ for $c, \beta \in \mathbb{R}$ and $k \in \mathbb{Z}$. They also studied the concept from the perspective of differential equations, motivating the term "Sturmian sequences." Since that time there has been tremendous progress in the study of such sequences from the perspective of Combinatorics on Words (Lothaire, 2001). However, the recent (and highly-approachable) paper of Rajpathak, Pillai and Bandyopadhyay (Chaos, Vol. 22, 2012) on the piecewise-linear map-with-a-gap discovers such sequences without recognising them as mechanical sequences. Proposition 9 of this section is a substantial generalisation of that result and we could not find this proposition explicitly stated in the literature. Our result is not surprising if one has the intuition that there is a topological conjugacy between the maps of this section and the piecewise linear map-with-a-gap. However, it might be difficult to explicitly identify the appropriate topological conjugacy and thereby prove our result for all cases considered here.

### 2.1 Definitions

Let $\pi$ denote a word consisting of a string of 0s and 1s in which the $k^{th}$ letter is $\pi_k$ and letters $i, i+1, \ldots, j$ are $\pi_{i:j}$. Let $|\pi|$ be the length of $\pi$ and $|\pi|_w$ for a word $w$ be the number of times that word $w$ appears in $\pi$. Let $\epsilon$ denote the empty word and $\pi^\omega$ denote the infinite word constructed by repeatedly concatenating $\pi$.

Consider two functions $\phi_0 : \mathcal{I} \to \mathcal{I}$ and $\phi_1 : \mathcal{I} \to \mathcal{I}$ where $\mathcal{I}$ is an interval of $\mathbb{R}$. We define the transformation $\phi_\pi : \mathcal{I} \to \mathcal{I}$ for any word $\pi$ by the composition

$$\phi_\pi(x) := \phi_{\pi_{|\pi|}} \circ \cdots \circ \phi_{\pi_2} \circ \phi_{\pi_1}(x).$$

Let $y_\pi \in \mathcal{I}$ be the fixed point of $\phi_\pi$, so $\phi_\pi(y_\pi) = y_\pi$, assuming a unique fixed point on $\mathcal{I}$ exists.

Given $x \in \mathcal{I}$, we call the sequence $(x_k : k \geq 1)$ the $x$-*threshold orbit for* $\phi_0, \phi_1$ if

$$x_1 = \phi_1(x), \qquad x_{k+1} = \begin{cases} \phi_1(x_k) & \text{if } x_k \geq x \\ \phi_0(x_k) & \text{if } x_k < x \end{cases} \qquad \text{for } k \geq 1.$$

We call $\pi$ the $x$-*threshold word for* $\phi_0, \phi_1$ if it is the shortest word such that $x_{k+1} = \phi_{(\pi^\omega)_k}(x_k)$ for all $k \geq 1$. We shall just write $x$-*threshold orbit* and $x$-*threshold word* where $\phi_0, \phi_1$ are obvious from the context.

For $p \geq 1$, let $L_p, R_p$ be the morphisms (substitutions)

$$L_p : \begin{cases} 0 \to 0^{p+1}1 \\ 1 \to 0^p 1 \end{cases} \qquad\qquad R_p : \begin{cases} 0 \to 01^p \\ 1 \to 01^{p+1} \end{cases} .$$

We say $\pi$ is a *valid word* if $\pi \in \{0, 1\}$ or $\pi \in \{L_p(w), R_p(w) : p \geq 1\}$ for some valid word $w$.

**Remark.** The morphisms $L_p, R_p$ generate the Christoffel tree so *valid words are mechanical words*. To see this, note that the Christoffel tree is generated by the following morphisms (Berstel *et al*, 2008, p. 37)

$$G : \begin{cases} 0 \to 0 \\ 1 \to 01 \end{cases} \qquad\qquad \tilde{D} : \begin{cases} 0 \to 01 \\ 1 \to 1 \end{cases} .$$

We may translate (from English to French) as $L_p = G^p \circ \tilde{D}$ and $R_p = \tilde{D}^p \circ G$ so any composition of $L_p$ and $R_p$ can be written as a composition of $G$ and $\tilde{D}$. Likewise, any composition of $G$ and $\tilde{D}$ can be written as a composition of $L_p$ and $R_p$. Specifically if $p_k, q_k, p_{k+1} \geq 2$ then

$$\cdots \circ G^{p_k - 1} \circ \tilde{D}^{q_k} \circ G^{p_{k+1}} \circ \tilde{D} \circ \cdots$$
$$= \cdots \circ (G^{p_k - 1} \circ \tilde{D}) \circ (\tilde{D}^{q_k - 1} \circ G) \circ (G^{p_{k+1} - 1} \circ \tilde{D}) \circ \cdots$$
$$= \cdots \circ L_{p_k - 1} \circ R_{q_k - 1} \circ L_{p_{k+1} - 1} \circ \cdots$$

whereas if $q_k = 1$ we have

$$\cdots \circ G^{p_k - 1} \circ \tilde{D} \circ G^{p_{k+1}} \circ \tilde{D} \circ \cdots$$
$$= \cdots \circ (G^{p_k - 1} \circ \tilde{D}) \circ (G^{p_{k+1}} \circ \tilde{D}) \circ \cdots$$
$$= \cdots \circ L_{p_k - 1} \circ L_{p_{k+1}} \circ \cdots .$$

A symmetric argument holds if $p_k = 1$ or $p_{k+1} = 1$.

## 2.2 Fixed Points

Throughout, we make the following assumption about $\phi_0, \phi_1$. The existence of fixed points $y_0, y_1$ is addressed immediately thereafter.

**Assumption A2.** *Functions* $\phi_0 : \mathcal{I} \to \mathcal{I}, \phi_1 : \mathcal{I} \to \mathcal{I}$, *where $\mathcal{I}$ is an interval of $\mathbb{R}$, are increasing and non-expansive. Equivalently, for all $x, y \in \mathcal{I} : x < y$ and for $k \in \{0, 1\}$ we have*

$$\underbrace{\phi_k(x) < \phi_k(y)}_{\text{increasing}} \qquad \text{and} \qquad \underbrace{\phi_k(y) - \phi_k(x) < y - x}_{\text{non-expansive}} .$$

*Furthermore, the fixed points $y_0, y_1$ of $\phi_0, \phi_1$ satisfy $y_1 < y_0$.*

**Proposition 1.** *Suppose A2 holds, that $x \in \mathcal{I}$ and that $w$ is any non-empty word. Then $\phi_w(x)$ is increasing and non-expansive. Further, the fixed point $y_w$ exists and is unique.*

*Proof.* First we show that $\phi_w(x)$ is increasing, by induction. In the base case, $|w| = 1$ and the claim follows from A2. For the inductive step assume $\phi_u(x)$ is increasing, where $w = au$ for some $a \in \{0, 1\}$ and word $u$. Then for any $x, y \in \mathcal{I} : x < y$,

$$\phi_w(y) = \phi_u(\phi_a(y))$$
$$> \phi_u(\phi_a(x)) \qquad \text{as } \phi_a(y) > \phi_a(x) \text{ and } \phi_u \text{ is increasing}$$
$$= \phi_w(x).$$

Therefore $\phi_w$ is increasing.

Now we show that $\phi_w(x)$ is non-expansive, by induction. If $|w| = 1$ then this follows from A2. Else, say $\phi_u(x)$ is non-expansive where $w = ua$ and $a \in \{0, 1\}$. Then for any $x, y \in \mathcal{I} : x < y$,

$$
\begin{aligned}
\phi_w(y) - \phi_w(x) &= \phi_a(\phi_u(y)) - \phi_a(\phi_u(x)) \\
&< \phi_u(y) - \phi_u(x) && \text{as } \phi_u(y) > \phi_u(x) \text{ and } \phi_a \text{ is non-expansive} \\
&< y - x && \text{as } \phi_u \text{ is non-expansive.}
\end{aligned}
$$

Therefore $\phi_w$ is non-expansive.

Let $\psi(x) := \max\{\phi_0(x), \phi_1(x)\}$. As $\phi_1$ is non-expansive we have

$$y_1 = \phi_1(y_1) > \phi_1(y_0) + y_1 - y_0$$

which rearranges to give $\phi_1(y_0) < y_0$, so that $\psi(y_0) = y_0$. Also $\psi$ is increasing as $\phi_0, \phi_1$ are increasing, so $\phi_w(y_0) \leq \psi^{(|w|)}(y_0) = y_0$.

We now prove that $y_w$ exists. The argument of the previous paragraph shows that $g(x) := x - \phi_w(x)$ satisfies $g(y_0) \geq 0$. A symmetric argument leads to the conclusion that $g(y_1) \leq 0$. Clearly $g(x)$ is a continuous function, so by the intermediate value theorem, there is some $y \in [y_0, y_1]$ for which $g(y) = 0$. Equivalently $y = \phi_w(y)$. Therefore a fixed point $y_w$ exists.

To show that the fixed point is unique, suppose both $y$ and $z$ are fixed points with $y > z$. As $\phi_w$ is non-expansive we have $\frac{\phi_w(y) - \phi_w(z)}{y - z} < 1$. Yet, as $\phi_w(y) = y, \phi_w(z) = z$ we have

$$\frac{\phi_w(y) - \phi_w(z)}{y - z} = 1.$$

This is a contradiction. Therefore the fixed point is unique. $\qquad\square$

Given a word $w$, the next proposition shows when the transformation $\phi_w$ increases or decreases its argument and what might be deduced from such an increase or decrease.

**Proposition 2.** *Suppose A2 holds, $x \in \mathcal{I}$ and $w$ is any non-empty word. Then*

$$x < \phi_w(x) \Leftrightarrow \phi_w(x) < y_w \Leftrightarrow x < y_w \quad \text{and} \quad x > \phi_w(x) \Leftrightarrow \phi_w(x) > y_w \Leftrightarrow x > y_w.$$

*Proof.* We use Proposition 1 throughout the argument without further mention.

Say $x < y_w$. As $\phi_w$ is increasing,

$$\phi_w(x) < \phi_w(y_w) = y_w$$

where the equality is the definition of $y_w$. Also, as $\phi_w$ is non-expansive,

$$y_w = \phi_w(y_w) < \phi_w(x) + y_w - x$$

which rearranges to give $x < \phi_w(x)$.

Now say $x > y_w$. As above, we then have $\phi_w(x) > \phi_w(y_w) = y_w$ and

$$y_w = \phi_w(y_w) > \phi_w(x) + y_w - x$$

so that $x > \phi_w(x)$.

The contrapositive of $x > y_w \Rightarrow \phi_w(x) > y_w$ is $\phi_w(x) \leq y_w \Rightarrow x \leq y_w$. But if $\phi_w(x) \neq y_w$ then $x \neq y_w$ as $\phi_w$ is increasing and therefore injective. Thus $\phi_w(x) < y_w \Rightarrow x < y_w$.

The contrapositive of $x > y_w \Rightarrow x > \phi_w(x)$ is $x \leq \phi_w(x) \Rightarrow x \leq y_w$. But if $x \neq \phi_w(x)$ then $x \neq y_w$ as $y_w$ is a fixed point. So we can conclude that $x < \phi_w(x) \Rightarrow x < y_w$.

By symmetry, $\phi_w(x) > y_w \Rightarrow x > y_w$ and $x > \phi_w(x) \Rightarrow x > y_w$. This completes the proof. $\quad\square$

**Proposition 3.** *Suppose A2 holds and $\pi$ is any word satisfying $|\pi|_0\, |\pi|_1 > 0$. Then $y_1 < y_\pi < y_0$.*

*Proof.* Say $y_\pi \leq y_1$. As $|\pi|_0 > 0$ we can write $\pi =: s01^q$ for some $q \geq 0$. Thus

$$
\begin{aligned}
y_\pi = \phi_\pi(y_\pi) &\leq \phi_{s01^q}(y_1) & \text{as } \phi_\pi \text{ is increasing} \\
&= \phi_{s0}(y_1) & \text{as } \phi_\epsilon(y_1) = \phi_1(y_1) = y_1 \\
&> \phi_s(y_1) & \text{by Proposition 2} \\
&\geq y_1 & \text{by repeating the same argument if } |s|_0 > 0.
\end{aligned}
$$

But this contradicts $y_\pi \leq y_1$. Therefore $y_\pi > y_1$.

A symmetrical argument leads to the conclusion that $y_\pi < y_0$. □

**Proposition 4.** *If A2 holds and $n \geq 1$ then $y_{10^{n-1}} < y_{010^{n-1}} < y_{10^n}$ and $y_{01^n} < y_{101^{n-1}} < y_{01^{n-1}}$.*

*Proof.* As $y_{10^{n-1}} < y_0$ by Proposition 3 we have $\phi_0(y_{10^{n-1}}) > y_{10^{n-1}}$ by Proposition 2 so that

$$\phi_{010^{n-1}}(y_{10^{n-1}}) = \phi_{10^{n-1}}(\phi_0(y_{10^{n-1}})) > \phi_{10^{n-1}}(y_{10^{n-1}}) = y_{10^{n-1}}$$

so Proposition 2 gives $y_{010^{n-1}} > y_{10^{n-1}}$.

Furthermore $y_{10^n} = \phi_0(y_{010^{n-1}})$ by definition of $y_\pi$ and $y_{010^{n-1}} < y_0$ by Proposition 3 so that $\phi_0(y_{010^{n-1}}) > y_{010^{n-1}}$ by Proposition 2. Thus $y_{10^n} > y_{010^{n-1}}$.

The proof that $y_{01^n} < y_{101^{n-1}} < y_{01^{n-1}}$ is symmetrical. □

**Proposition 5.** *Suppose A2 holds, $M \in \{L_q, R_q : q \geq 1\}$ and $\tilde{w}$ is any word. Let $\tilde{y}_v$ be the fixed point of $\tilde{\phi}_v := \phi_{M(v)}$ for any word $v$ and let $0w1 := M(0\tilde{w}1)$. Then*

$$\tilde{x} \in [\tilde{y}_{01\tilde{w}}, \tilde{y}_{10\tilde{w}}] \Leftrightarrow x := \phi_{0^q}(\tilde{x}) \in [y_{01w}, y_{10w}].$$

*Proof.* Say $M = L_q$. Note that

$$
\begin{aligned}
\phi_{0^q}(\tilde{y}_{01\tilde{w}}) &= \phi_{0^q}(y_{L_q(01\tilde{w})}) & \text{as } \tilde{y}_v \text{ is the fixed point of } \tilde{\phi}_v = \phi_{L_q(v)} \\
&= \phi_{0^q}(y_{0^q01L_q(1\tilde{w})}) & \text{as } L_q(0) = 0^q01 \\
&= y_{01L_q(1\tilde{w})0^q} & \text{as } \phi_a(y_{ab}) = y_{ba} \text{ for any words } a, b \\
&= y_{01w} & \text{as } 0w1 = L_q(0\tilde{w}1) = 0L_q(1\tilde{w})0^q1
\end{aligned}
$$

and

$$
\begin{aligned}
\phi_{0^q}(\tilde{y}_{10\tilde{w}}) &= \phi_{0^q}(y_{L_q(10\tilde{w})}) \\
&= \phi_{0^q}(y_{0^q1L_q(0\tilde{w})}) \\
&= y_{1L_q(0\tilde{w})0^q} \\
&= y_{10w} & \text{as } 0w1 = L_q(0\tilde{w})0^q1.
\end{aligned}
$$

Proposition 1 shows that $\tilde{y}_{01\tilde{w}}, \tilde{y}_{10\tilde{w}}$ exist. So the above equalities show that an inverse $\phi_{0^q}^{(-1)}(x)$ exists for $x \in \{y_{01w}, y_{10w}\}$. As $\phi_{0^q}$ is increasing and continuous, we have

$$x \in [y_{01w}, y_{10w}] \Leftrightarrow \tilde{x} \in [\phi_{0^q}^{(-1)}(y_{01w}), \phi_{0^q}^{(-1)}((y_{10w})] = [\tilde{y}_{01\tilde{w}}, \tilde{y}_{10\tilde{w}}].$$

The proof for $M = R_q$ is symmetric. □

### 2.3  $x$-Threshold Words

**Proposition 6.** *Suppose A2 holds, $\pi$ is the $x$-threshold word and $n \geq 1$. Then*

1. $x \leq y_{10^{n-1}} \Rightarrow |\pi^\omega|_{0^n} = 0$

2. $x \geq y_{010^{n-1}} \Rightarrow |\pi^\omega|_{10^{n-1}1} = 0$

3. $x \geq y_{01^{n-1}} \Rightarrow |\pi^\omega|_{1^n} = 0$

4. $x \leq y_{101^{n-1}} \Rightarrow |\pi^\omega|_{01^{n-1}0} = 0$

*Proof.* If $x \leq y_1$ then it follows from Proposition 2 that the $x$-threshold word is $\pi = 1$. Likewise if $x > y_0$ then the $x$-threshold word is $\pi = 0$. In these cases Claims 1 and 2 hold, so in the following we assume that $y_1 < x \leq y_0$.

*Claim 1:* Let $(x_k)$ the $x$-threshold orbit. If $(\pi^\omega)_{k:k+n-2} = 0^{n-1}$ for some $k$, then

$$
\begin{aligned}
x_{k+n-1} &= \phi_{0^{n-1}}(x_k) & \text{by definition of } (x_k) \\
&\geq \phi_{0^{n-1}}(\phi_1(x)) & \text{as } x_k \geq \phi_1(x) \text{ for all } k \geq 0 \text{ and } \phi_{0^{n-1}} \text{ is increasing} \\
&= \phi_{10^{n-1}}(x) \\
&\geq x & \text{if } x \leq y_{10^{n-1}} \text{ by Proposition 2.}
\end{aligned}
$$

But if $x_{k+n-1} \geq x$ then $\pi_{k+n-1} = 1$ by definition $\pi$. Therefore $|\pi|_{0^n} = 0$.

*Claim 2:* Let $(x_k)$ be the $x$-threshold orbit. If $(\pi^\omega)_{k:k+n-1} = 10^{n-1}$ for some $k$, then

$$
\begin{aligned}
x_{k+n} &= \phi_{10^{n-1}}(x_k) \\
&< \phi_{10^{n-1}}(\phi_0(x)) & \text{as } x_k < \phi_0(x) \text{ for all } k \geq 0 \text{ and } \phi_{10^{n-1}} \text{ is increasing} \\
&= \phi_{010^{n-1}}(x) \\
&\leq x & \text{if } x \geq y_{010^{n-1}} \text{ by Proposition 2.}
\end{aligned}
$$

But if $x_{k+n} < x$ then $(\pi^\omega)_{k+n} = 0$. Therefore $|\pi|_{10^{n-1}1} = 0$.

The proof of Claims 3 and 4 is symmetrical. $\qquad\square$

**Proposition 7.** *Suppose A2 holds and $\pi$ is a $x$-threshold word. Then*

1. $|\pi|_{00} > 0 \Rightarrow \pi = L_n(w)$ *for some word $w$ and some $n \geq 1$*

2. $|\pi|_{11} > 0 \Rightarrow \pi = R_n(w)$ *for some word $w$ and some $n \geq 1$*

*Proof.* First, applying Claims 1 and 3 of Proposition 6 with $n = 2$ we have $|\pi|_{00} = 0$ for $x \leq y_{10}$ and $|\pi|_{11} = 0$ for $x \geq y_{01}$. Furthermore $y_{10} = \phi_0(y_{01}) > y_{01}$ by Proposition 2. Thus $\pi$ cannot contain both 00 and 11.

So, if $|\pi|_{00} > 0$ then $\pi$ is of the form $0^{q_1}10^{q_2}1\ldots$ with strings of 0s separated by individual 1s. Let $q := \min_k q_k$. By Propositions 4 and 6, $I_q := (y_{10^{q-1}}, y_{010^q})$ is the only set of $x$ values for which $\pi^\omega$ can contain $10^q1$. Thus $\pi^\omega$ can only contain both $10^q1$ and $10^{q+1}1$ in the interval

$$ F_q := I_q \cap I_{q+1} = (y_{10^{q-1}}, y_{010^q}) \cap (y_{10^q}, y_{010^{q+1}}) = (y_{10^q}, y_{010^q}) $$

noting Proposition 4 gives $y_{10^{q-1}} < y_{010^{q-1}} < y_{10^q} < y_{010^q}$.

Finally, we have $F_q \cap F_{q'} = \emptyset$ for $q \neq q'$, which also follows from Proposition 4. Thus if $|\pi|_{00} > 0$ then $\pi$ is a concatenation of $L_q(0)$ and $L_q(1)$. Equivalently $\pi = L_q(w)$ for some word $w$ and some $q \geq 1$ as in Claim 1.

The proof of Claim 2 is symmetric. $\qquad\square$

**Proposition 8.** *Suppose A2 holds and $\pi$ is a $x$-threshold word. Then $\pi$ is a valid word.*

*Proof.* There are three cases to consider: either $|\pi|_{00} = |\pi|_{11} = 0$ or $|\pi|_{00} > 0$ or $|\pi|_{11} > 0$.

*First case:* The only non-empty words not containing 00 or 11 are $0, 1, (01)^n, (10)^n$ for some $n \geq 1$. Now $x$-threshold words start with 0 unless $x \leq y_1$ (in which case $\pi = 1$) so $\pi \neq (10)^n$. Further, the $x$-threshold word was defined to be the shortest word such that such that $x_{k+1} = A_{(\pi^\omega)_k} x_k$ so this leaves us with the options $0, 1, 01$. These are all valid words.

*Second case:* If $\pi$ contains 00, we may write $\pi = L_q(w)$ for some word $w$, by Proposition 7. Now from point $x_k$ on the $x$-threshold orbit we have $\pi_{k:k+q} = 0^{q+1}$ if and only if $\phi_{0^q}(x_k) < x$ which

corresponds to $x_k < \phi_0^{(-q)}(x) =: \tilde{x}$. So the word $w$ corresponds to a $\tilde{x}$-threshold orbit $(\tilde{x}_k : k \geq 1)$ for $\psi_0(x) := \phi_{0^{q+1}1}(x), \psi_1(x) := \phi_{0^q1}(x)$. To spell it out, we have

$$\tilde{x}_1 = \psi_1(\tilde{x}), \qquad \tilde{x}_{k+1} = \psi_{w_k}(\tilde{x}_k), \qquad w_k = \begin{cases} 1 & \text{if } \tilde{x}_k \geq \tilde{x} \\ 0 & \text{if } \tilde{x}_k < \tilde{x} \end{cases} \qquad \text{for } k \geq 1$$

and as for the original system, we define $\tilde{y}_\pi$ as the fixed point $\tilde{y}_\pi = \psi_\pi(\tilde{y}_\pi)$.

Now $\psi_0, \psi_1$ are non-negative, as $\phi_0, \phi_1$ are non-negative. Also $\psi_0, \psi_1$ are monotonically increasing and non-expansive by Proposition 1. Further,

$$\phi_{0^{q+1}1}(y_{0^q1}) = \phi_{0^q1}(\phi_0(y_{0^q1})) > \phi_{0^q1}(y_{0^q1}) = y_{0^q1}$$

so that $y_{0^{q+1}1} > y_{0^q1}$ by Proposition 2. But by definition $\tilde{y}_0 = y_{0^{q+1}1}$ and $\tilde{y}_0 = y_{0^q1}$, so that $\tilde{y}_1 < \tilde{y}_0$. Therefore $\psi_0, \psi_1$ satisfy A2.

*Third case:* We prove that $\pi = R_q(w)$ for some positive integer $q$ and word $w$. We also show that word $w$ is a $\hat{x}$-threshold word for a pair of functions (say) $\chi_0, \chi_1$ which satisfy A2. The argument is symmetric to the second case, so it is omitted.

In conclusion, either

1. $\pi \in \{0, 1, L_1(1)\}$ which are valid words

2. $\pi = L_q(w)$ where $w$ is a $\tilde{x}$-threshold word for $\psi_0, \psi_1$ which satisfy Propositions 1-7 and therefore $w$ satisfies this conclusion

3. or $\pi = R_q(w)$ where $w$ is a $\hat{x}$-threshold word for $\chi_0, \chi_1$ which satisfy Propositions 1-7 and therefore $w$ satisfies this conclusion.

Thus $\pi$ is a valid word. This completes the proof. $\qquad\square$

The following proposition shows that all valid words are $x$-threshold words and tells us explicitly which values of $x$ produce a given valid word. It is one of the key results of the main paper.

**Proposition 9.** *Suppose A2 is satisfied and $0w1$ is any valid word. Then*

$$0w1 \text{ is the } x\text{-threshold word} \iff x \in [y_{01w}, y_{10w}].$$

*Proof.* Let $V_1 := \{L_q(1), R_q(1) : q \geq 1\}, V_{n+1} := \{L_q(v), R_q(v) : v \in V_n, q \geq 1\}$. Note that $V_1$ contains $L_q(0) = 0^{q+1}1 = L_{q+1}(1)$ and $R_q(0) = 01^q$ which for $q \geq 2$ equals $R_{q-1}(1)$ and for $q = 1$ equals $01 = L_1(1)$. Thus $\cup_{n=1}^\infty V_n$ is the set of all valid words of form $0w1$.

We use induction with hypothesis

$$H_n : \quad 0w1 \in V_n \text{ is the } x\text{-threshold word} \iff x \in [y_{01w}, y_{10w}]$$

*Base case ($H_1$).* Say $0w1 = 0^q1$ is the $x$-threshold word. Then

$$x > \phi_{(10^q)^n 10^{q-1}}(x) \qquad\qquad \text{for all } n \geq 0$$
$$= \phi_{(010^{q-1})^n}(\phi_{10^{q-1}}(x))$$
$$\Rightarrow x \geq \lim_{n\to\infty} \phi_{(010^{q-1})^n}(\phi_{10^{q-1}}(x)) = y_{010^{q-1}}.$$

The definition of the $x$-threshold word also gives $x \leq \phi_{10^q}(x)$. Therefore $x \geq y_{10^q}$ by Proposition 2. Thus if $0^q1$ is the $x$-threshold word then $x \in [y_{01w}, y_{10w}]$.

Now say $x \in [y_{010^{q-1}}, y_{10^q}]$. Proposition 3 gives $y_0 < x < y_1$ so that the $x$-threshold orbit $(x_k)$ is contained in $(y_0, y_1)$. So Proposition 2 shows that $\phi_0(x_k) > x_k$ and $\phi_1(x_k) < x_k$ for all $k \geq 0$. So to prove that the $x$-threshold word is $0^q1$ we need only show that $\phi_{(10^q)^n 10^{q-1}}(x) < x$ and $\phi_{(10^q)^n}(x) \geq x$ for all $n \geq 0$. But if $x \geq y_{010^{q-1}}$ then for all $n \geq 0$

$$x \geq \phi_{(010^{q-1})^n}(x) \qquad\qquad \text{by Proposition 2}$$
$$> \phi_{(010^{q-1})^n}(\phi_{10^{q-1}}(x)) \qquad \text{as } y_{10^{q-1}} < y_{010^{q-1}} \leq x \text{ by Claim 3 of Proposition 4}$$
$$= \phi_{(10^q)^n 10^{q-1}}(x).$$

Also if $x \leq y_{10^q}$ then $\phi_{(10^q)^n}(x) \geq x$ for all $n \geq 0$ by Proposition 2. Therefore for $0w1 = 0^q1$, we have $x \in [y_{01w}, y_{10w}]$ implies that $0w1$ is the $x$-threshold word.

For $0w1 = 01^q$, the proof that $\pi = 01^q \Leftrightarrow x \in [y_{01w}, y_{10w}]$ is symmetric, so it is omitted.

*Inductive Step.* Assume $0\tilde{w}1$ satisfies $H_n$.

Say $0w1 = L_q(0\tilde{w}1)$. Let $k_i := |L_q(((0\tilde{w}1)^\omega)_{1:i-1})| + 1$ so $(\pi^\omega)_{k_i}$ is aligned with the start of the $i^{th}$ letter of $(0\tilde{w}1)^\omega$. Let $x_k := \phi_{((10w)^\omega)_{1:k}}(x), \tilde{x}_i := x_{k_i}, x = \phi_{0^q}(\tilde{x})$ and let $\tilde{y}_v$ denote the fixed point of $\tilde{\phi}_v := \phi_{L_q(v)}$ for any word $v$. Then we have

$$L_q(0\tilde{w}1) \text{ is the } x\text{-threshold word for } \phi_0, \phi_1$$
$$\Leftrightarrow \quad ((0w1)^\omega)_{k_i:k_i+q} = 0^{q+1} \text{ if and only if } \phi_{0^q}(x_{k_i}) < x$$
$$\Leftrightarrow \quad ((0\tilde{w}1)^\omega)_i = 0 \text{ if and only if } \tilde{x}_i < \tilde{x}$$
$$\Leftrightarrow \quad 0\tilde{w}1 \text{ is the } \tilde{x}\text{-threshold word for } \tilde{\phi}_0, \tilde{\phi}_1$$
$$\Leftrightarrow \quad \tilde{x} \in [\tilde{y}_{01\tilde{w}}, \tilde{y}_{10\tilde{w}}] \text{ as } 0\tilde{w}1 \text{ satisfies } H_n$$
$$\Leftrightarrow \quad x \in [y_{01w}, y_{10w}] \text{ by Proposition 5}$$

Symmetrically we may conclude that $\pi = 0w1 = R_q(0\tilde{w}1) \Leftrightarrow x \in [y_{01w}, y_{10w}]$. Therefore $H_{n+1}$ is true.

This completes the proof. $\qquad\qquad\qquad\qquad\qquad\qquad\qquad\qquad\qquad\qquad\qquad\qquad\quad$ □

## 3   Continuity of the Index

We showed that the Whittle index is increasing on the domain of each fixed Christoffel word. However, we also need to show that the index is continuous as we move between words. So here we prove the following proposition.

**Proposition 10.** *Suppose $\lambda(\cdot)$ is as in the main paper. Then $\lambda(x)$ is a continuous function of $x \in \mathbb{R}_+$.*

We use the following definitions.

**Definition.** Let $\tilde{w}$ be the reverse of word $w$, $w^\omega$ be the word constructed by concatenating $w$ infinitely many times, $|w|$ be the length of word $w$ and $|w|_u$ be the number of times that word $u$ is a factor of $w$.

**Definition.** For a possibly-infinite word $w$ and numbers $x \in \mathbb{R}, \beta \in (0,1)$ define

$$S(w, x) := \sum_{n=0}^{|w|-1} \beta^n \phi_{w_{1:n}}(x)$$

$$\lambda(0w1, x) := \frac{1 - \beta^{|0w1|}}{1 - \beta} \left( S((01w)^\omega, x) - S((10w)^\omega, x) \right).$$

**Remark.** If $\pi$ is the $x$-threshold word then $\lambda(x) = \lambda(\pi, x)$ where $\lambda(x)$ is the Whittle index.

**Remark.** For a word $ab$, this definition gives

$$S(ab, x) = S(a, x) + \beta^{|a|} S(b, \phi_a(x)) \tag{1}$$

so for $|\phi_{a^\omega}(x)| < \infty$ and $\beta \in (0,1)$ we have

$$S(a^\omega b, x) = S(a^\omega, x). \tag{2}$$

Further, if $x_a = \phi_a(x_a)$ then the formula for the sum of a geometric progression gives

$$S(a^\omega, x_a) = \frac{S(a, x_a)}{1 - \beta^{|a|}}. \tag{3}$$

**Definition.** Let $X_\pi$ be the range of $x$ for which the $x$-threshold word is $\pi$.

The following construction is closely related to the beautiful *Christoffel tree* (Berstel *et al*, 2008).

**Definition.** Consider the mapping $C$ which takes a sequence of words and returns a sequence containing the original words mingled with the concatenation of neighbouring words as follows:

$$C((a, b, c, d, \ldots, x, y, z)) := (a, ab, b, bc, c, cd, d, \ldots, x, xy, y, yz, z).$$

Now consider the sequences $t_k := C^{(k)}((0, 1))$ for $k \geq 0$. The first few such sequences are

$$
\begin{aligned}
t_0 &= (0, && && && && && && && && 1) \\
t_1 &= (0, && && && && 01, && && && && 1) \\
t_2 &= (0, && && 001, && && 01, && && 011, && && 1) \\
t_3 &= (0, && 0001, && 001, && 00101, && 01, && 01011, && 011, && 0111, && 1).
\end{aligned}
$$

**Remark.** If $u \in t_k$ then $|u| \geq 1$ for any $k \geq 0$. Now suppose $u, v$ are adjacent in $t_k$ and we have $|uv| \geq k + 2$. Then $t_{k+1}$ contains $u, uv, v$ from which we can construct $uuv$ and $uvv$. But $|uuv| = |u| + |uv| \geq 1 + k + 2 = k + 3$ and $|uvv| = |uv| + |v| \geq k + 2 + 1 = k + 3$. Thus, by induction, we have shown that

$$|uv| \geq k + 2 \qquad\qquad \text{for any adjacent pair } u, v \text{ in } t_k \text{ and any } k \geq 0. \qquad (4)$$

## 3.1 Long Common Prefixes

We gather the results needed to prove Proposition 10. Most of these results these relate to the notion that if $|x - y|$ is small and $a, b$ are the $x$- and $y$-threshold words, then words $a, b$ usually have a long common prefix, although this is not always the case.

The following simple result is repeatedly used in the other Lemmas of this subsection.

**Lemma 1.** *Suppose* $(0a1, 0b1)$ *is a standard pair. Then* $a10b = b01a$.

*Proof.* As $(0a1, 0b1)$ is a standard pair, $0a10b1 =: 0w1$ is a Christoffel word. As $0a1, 0b1, 0w1$ are Christoffel words, $a, b, w$ are palindromes. Thus $a10b = w = \tilde{w} = \tilde{b}01\tilde{a} = b01a$. $\square$

If $(0a1, 0b1)$ is a standard pair, then the interval $X_{0b1}$ is immediately to the left of $X_{0a1(0b1)^\omega}$. Since the words $0b1$ and $0a1(0b1)^\omega$ can differ within the first few letters, continuity of $\lambda(x)$ at $x = \sup X_{0b1}$ is not obvious. Similarly, $X_{(0a1)^\omega 0b1}$ is immediately to the left of $X_{0a1}$. However, the factors $1 - \beta^{|(0a1)^\omega 0b1|}$ and $1 - \beta^{|0a1|}$ appearing in the definitions of the corresponding Whittle indices are different for $|a| < \infty$. Thus continuity of $\lambda(x)$ at $x = \sup X_{0a1}$ is not obvious. The next two Lemmas address these questions.

**Lemma 2.** *Suppose* $(0a1, 0b1)$ *is a standard pair and let* $x = \phi_{10b}(x)$. *Then*

$$\lambda(0b1, x) = \lambda(0a1(0b1)^\omega, x).$$

*Proof.* The right-hand side $\lambda(0a1(0b1)^\omega, x)$ involves the sum

$$
\begin{aligned}
S(10a1(0b1)^\omega, x) &= S(10b01a(10b)^\omega, x) && \text{by Lemma 1} \\
&= S(10b, x) + \beta^{|10b|} S(01a(10b)^\omega, \phi_{10b}(x)) && \text{by 1} \\
&= S(10b, x) + \beta^{|10b|} S(01a(10b)^\omega, x) && \text{as } x = \phi_{10b}(x) \\
&= (1 - \beta^{|10b|}) S((10b)^\omega, x) + \beta^{|10b|} S(01a(10b)^\omega, x) && \text{by 3.} \quad (5)
\end{aligned}
$$

Now we note that repeated application of Lemma 1 gives

$$01a(10b)^\omega = 01a10b(10b)^\omega = 01b\,01a(10b)^\omega = (01b)^\omega 01a. \qquad (6)$$

Thus

$$\lambda(0a1(0b1)^\omega, x) = \frac{1 - \beta^{|0a1(0b1)^\omega|}}{1 - \beta} \left( S((01a1(0b1)^\omega)^\omega, x) - S((10a1(0b1)^\omega)^\omega, x) \right)$$

$$= \frac{S(01a1(0b1)^\omega, x) - S(10a1(0b1)^\omega, x)}{1 - \beta} \qquad \text{by 2}$$

$$= \frac{1 - \beta^{|10b|}}{1 - \beta} \left( S(01a(10b)^\omega, x) - S((10b)^\omega, x) \right) \qquad \text{by 5}$$

$$= \frac{1 - \beta^{|10b|}}{1 - \beta} \left( S((01b)^\omega, x) - S((10b)^\omega, x) \right) \qquad \text{by 6}$$

$$= \lambda(0b1, x).$$

This completes the proof. $\qquad\square$

**Lemma 3.** *Suppose* $(0a1, 0b1)$ *is a standard pair and let* $x = \phi_{01a}(x)$. *Then*

$$\lambda((0a1)^\omega 0b1, x) = \lambda(0a1, x).$$

*Proof.* The left-hand side $\lambda((0a1)^\omega 0b1, x)$ involves the sum

$$S(01(a10)^\omega 0b1, x) = S(01(a10)^\omega, x) \qquad \text{by 2}$$

$$= S(01a, x) + \beta^{|01a|} S((10a)^\omega, \phi_{01a}(x)) \qquad \text{by 1}$$

$$= S(01a, x) + \beta^{|01a|} S((10a)^\omega, x) \qquad \text{as } x = \phi_{01a}(x)$$

$$= (1 - \beta^{|01a|}) S((01a)^\omega, x) + \beta^{|01a|} S((10a)^\omega, x) \qquad \text{by 3.} \quad (7)$$

Thus

$$\lambda((0a1)^\omega 0b1, x) = \frac{1 - \beta^{|(0a1)^\omega 0b1|}}{1 - \beta} \left( S((01(a10)^\omega 0b1)^\omega, x) - S((10(a10)^\omega 0b1)^\omega, x) \right)$$

$$= \frac{1}{1 - \beta} \left( S(01(a10)^\omega 0b1, x) - S((10a)^\omega, x) \right) \qquad \text{by 2}$$

$$= \frac{1 - \beta^{|01a|}}{1 - \beta} \left( S((01a)^\omega, x) - S((10a)^\omega, x) \right) \qquad \text{by 7}$$

$$= \lambda(0a1, x).$$

This completes the proof. $\qquad\square$

To demonstrate continuity at other points, we will need to rely on the fact that nearby words often have a long common prefix as shown by the following two Lemmas.

**Lemma 4.** *Suppose* $(0a1, 0b1)$ *is a subsequence of* $t_k$ *for some* $k \geq 1$. *Then* $0b01a$ *is a prefix of both* $(0a1)^\omega$ *and* $0b(01b)^\omega$.

*Proof.* Let $a = b \cdots$ indicate that $b$ is a prefix of word $a$ and consider the statements

$$A(a, b) : (a10)^\omega = b \cdots \qquad \text{and} \qquad B(a, b) : (b01)^\omega = a \cdots .$$

It suffices to show that $A(a, b)$ and $B(a, b)$ are true for any adjacent words $0a1, 0b1$ in $t_k$ for $k \geq 0$. This is because

$$A(a, b) \Rightarrow (0a1)^\omega = 0a10(a10)^\omega = 0a10b \cdots = 0b01a \cdots$$

where the last equality follows from Lemma 1 and

$$B(a, b) \Rightarrow 0b(01b)^\omega = 0b01(b01)^\omega = 0b01a \cdots$$

which are the claims of the Lemma.

We shall use induction. Take $t_2 = (0, 001, 01, 011, 01)$ as the base case. We must show that $A(0, \epsilon), B(0, \epsilon), A(\epsilon, 1), B(\epsilon, 1)$ are true. However these statements are respectively that $(001)^\omega = \epsilon \cdots, (01)^\omega = 0 \cdots, (10)^\omega = 1 \cdots, (101)^\omega = \epsilon \cdots$ and are all true.

Otherwise, say $A(a, b), B(a, b)$ are true for any adjacency $0a1, 0b1$ in $t_k$. Let $0a10b1 = 0c1$ so

$$c = a10b = b01a$$

using Lemma 1 again. Then the statements $A(a, c), B(a, c), A(c, b), B(c, b)$ are all true as

$$(a10)^\omega = a10(a10)^\omega = a10b \cdots = c \cdots \qquad \text{by } A(a, b) \text{ and as } c = a10b$$
$$(c01)^\omega = c \cdots = a \cdots \qquad \text{as } c = a10b$$
$$(c10)^\omega = c \cdots = b \cdots \qquad \text{as } c = b01a$$
$$(b01)^\omega = b01(b01)^\omega = b01a \cdots = c \cdots \qquad \text{by } B(a, b) \text{ and as } c = b01a.$$

Thus $A(a, b), B(a, b)$ are true for all adjacencies $0a1, 0b1$ in $t_{k+1}$. This completes the proof. $\qquad \square$

**Lemma 5.** *Suppose $0a1, 0b1$ are adjacent in $t_k$ and that $0c1$ lies strictly between them in $t_{k'}$ for some $0 < k < k'$. Then $0c1 = 0b01a \cdots$.*

*Proof.* The interval of $t_{k'}$ between $0a1, 0b1$ is constructed from $0a1, 0b1$ in exactly the same way as $t_{k'-k}$ was constructed from $0, 1$. Thus $0c1 = (0a1)^q 0b1 \cdots$ for some positive integer $q$. Now recall that $0b01a = 0a10b$ by Lemma 1. Thus $0c1 = (0a1)^{q-1} 0a10b1 \cdots = (0a1)^{q-1} 0b01a1 \cdots = 0b(01a)^q 1 \cdots = 0b01a \cdots$ as claimed. $\qquad \square$

Although the existence of a long common prefix for nearby words suggests continuity, to prove anything we must bound the residual after removing the long common prefix. The following Lemma is one way to achieve this.

**Lemma 6.** *Suppose $x \geq y \geq 0$, let $0w1$ be the $x$-threshold word and let $(01w)^\omega = su, (10w)^\omega = s'u'$ where $|s| = |s'|$. Then $|S(u, \phi_s(y)) - S(u', \phi_{s'}(y))| \leq \frac{x+1}{1-\beta}$.*

*Proof.* The highest point on the orbits $(\phi_{((01w)^\omega)_{1:k}}(x) : k \geq 0)$ and $(\phi_{((10w)^\omega)_{1:k}}(x) : k \geq 0)$ is $x + 1$ since $0w1$ is the $x$-threshold word. The terms $a_k, b_k$ of the discounted sums

$$S(u, \phi_s(y)) =: \sum_{k=0}^\infty \beta^k a_k \text{ and } S(u', \phi_{s'}(y)) =: \sum_{k=0}^\infty \beta^k b_k$$

are from the orbits $(\phi_{((01w)^\omega)_{1:k}}(y) : k \geq 0)$ and $(\phi_{((10w)^\omega)_{1:k}}(y) : k \geq 0)$ and $\phi_{u''}(x) \geq \phi_{u''}(y)$ for any word $u''$ as $x \geq y$. Therefore terms $a_k, b_k$, are also no higher than $\phi_0(x) \leq x + 1$. Furthermore, terms $a_k, b_k$ are non-negative, so that $|a_k - b_k| \leq x + 1$. Thus $|S(u, \phi_s(y)) - S(u', \phi_{s'}(y))| \leq \sum_{k=0}^\infty \beta^k |a_k - b_k| \leq \sum_{k=0}^\infty \beta^k(x+1) = \frac{x+1}{1-\beta}$. $\qquad \square$

Although it is clear that $\lambda(\pi, x)$ is continuous, a bound on its slope is helpful.

**Lemma 7.** *Suppose $x \geq 0$ and that $0w1$ is a valid word. Then $|\lambda'(0w1, x)| \leq \frac{1}{(1-\beta)^2}$.*

*Proof.* The definition of $\lambda(0w1, x)$ gives

$$|\lambda'(0w1, x)| \leq \frac{1 - \beta^{|0w1|}}{1 - \beta} \sum_{k=0}^\infty \beta^k \left| \phi'_{((01w)^\omega)_{1:k}}(x) - \phi'_{((10w)^\omega)_{1:k}}(x) \right| \leq \frac{1}{1-\beta} \sum_{k=0}^\infty \beta^k = \frac{1}{(1-\beta)^2}$$

where the second inequality follows as $0 \leq \beta^{|0w1|} < 1$ and $0 \leq \phi'_u(x) \leq 1$ for any word $u$ since $0 \leq \phi'_1(x) \leq \phi'_0(x) \leq 1$. $\qquad \square$

We use one more result about $\phi_0, \phi_1$ of the main paper.

**Lemma 8.** *Suppose $\phi_0(x)$ and $\phi_1(x)$ are as in the main paper and $x \in \mathbb{R}_+$. Then $\phi_{01}(x) < \phi_{10}(x)$.*

*Proof.* The definitions of $\phi_0, \phi_1$ give

$$\phi_{10}(x) - \phi_{01}(x) =$$

$$(b-a)\frac{(ab+b+a)x^2 + (2ab+3b+3a+2)x + ab+2b+2a+3}{((ab+b+a)x+ab+b+2a+1)((ab+b+a)x+ab+2b+a+1)}$$

which is positive as $b > a$ and $x \geq 0$. $\qquad\square$

Our proof of continuity will rely on the standard $(\epsilon, \delta)$ definition in which we will put $\delta = l_k$ where $l_k$ is defined in the following Lemma.

**Lemma 9.** *For any $\epsilon > 0$ there is a $k < \infty$ such that $0 < l_k := \inf\{|X_\pi| : \pi \in t_k\} < \epsilon$.*

*Proof.* Say $0a1, 0b1$ are adjacent in $t_k$. Then by construction of $t_{k+i}$, the gap $(z_{10b}, z_{01a})$ contains $2^i - 1$ intervals corresponding to words of $t_{k+i} \backslash t_k$. Each of these intervals is at most $\frac{z_{01a} - z_{10b}}{2^i - 1}$ in length. Thus $\lim_{k\to\infty} l_k = 0$. This demonstrates the existence of a $k < \infty$ such that $l_k < \epsilon$.

To show that $l_k > 0$ for finite $k$, we shall demonstrate that assuming $l_k = 0$ leads to a contradiction. If $l_k = 0$ then there is some word $0w1 \in t_k$ such that $z_{10w} = z_{01w} =: x$. Therefore $\phi_{10w}(x) = \phi_{01w}(x)$. Now in $\mathbb{R}_+$, functions $\phi_0(x), \phi_1(x)$ have inverses, so $\phi_w^{-1}(x)$ is well-defined. Therefore

$$\phi_{10}(x) = \phi_w^{-1} \circ \phi_{10w}(x) = \phi_w^{-1} \circ \phi_{01w}(x) = \phi_{01}(x)$$

which contradicts Lemma 8 as $x \geq 0$. $\qquad\square$

## 3.2 Proof of Continuity

*Proof.* We wish to show that for any $\epsilon > 0$, there exists a $\delta > 0$ such that for any $|x - y| < \delta$ we have $\Delta := |\lambda(x) - \lambda(y)| < \epsilon$. Without loss of generality we assume that $x \geq y$.

Specifically, we shall put $\delta = l_k > 0$ where $l_k$ is as defined in Lemma 9 and $k$ is any positive integer such that $\frac{l_k}{(1-\beta)^2} < \frac{\epsilon}{2}$ and such that $2\frac{x+1}{(1-\beta)^2}\beta^{k+1} < \frac{\epsilon}{2}$. The existence of such a $k$ is guaranteed by Lemma 9 and because $\beta \in (0,1)$.

Let $0a1, 0b1$ be the $x$- and $y$-threshold words. If these words are the same then

$$\Delta = |\lambda(0a1, x) - \lambda(0a1, y)| \leq |y - x| \sup_{z \in [x,y]} |\lambda'(0a1, z)| \leq \frac{|y-x|}{(1-\beta)^2} \leq \frac{l_k}{(1-\beta)^2} < \frac{\epsilon}{2}$$

where the second inequality follows from Lemma 7, the third from $|y - x| < \delta = l_k$ and the fourth from the definition of $k$.

Otherwise $0a1 \neq 0b1$. In this case, let $(0e1, 0b1)$ be the standard pair for word $0b1$, let $\underline{a} = \phi_{10a}(\underline{a})$ and $\bar{b} = \phi_{01b}(\bar{b})$. Noting that $y \leq \bar{b} \leq \underline{a} \leq x$, our strategy is to write

$$\Delta = |\Delta_1 + \Delta_2 + \Delta_3 + \Delta_4 + \Delta_5 + \Delta_6|$$
$$\Delta_1 := \lambda(0b1, y) - \lambda(0b1, \bar{b})$$
$$\Delta_2 := \lambda(0b1, \bar{b}) - \lambda(0e1(0b1)^\omega, \bar{b})$$
$$\Delta_3 := \lambda(0e1(0b1)^\omega, \bar{b}) - \lambda((0a1)^\omega, \bar{b})$$
$$\Delta_4 := \lambda((0a1)^\omega, \bar{b}) - \lambda((0a1)^\omega, \underline{a})$$
$$\Delta_5 := \lambda((0a1)^\omega, \underline{a}) - \lambda(0a1, \underline{a})$$
$$\Delta_6 := \lambda(0a1, \underline{a}) - \lambda(0a1, x).$$

Lemma 7 and the choice of $\delta$ give

$$|\Delta_1| + |\Delta_4| + |\Delta_6| \leq \frac{\bar{b} - y + \underline{a} - \bar{b} + x - \underline{a}}{(1-\beta)^2} < \frac{l_k}{(1-\beta)^2} \leq \frac{\epsilon}{2} \tag{8}$$

while Lemmas 2 and 3 give

$$\Delta_2 = \Delta_5 = 0. \tag{9}$$

It remains to consider $\Delta_3$. It follows from the definition of $l_k$, that for some adjacent words $0c1, 0d1$ in $t_k$: either $0a1 = 0c1$ or $0a1$ is a word strictly between $0c1$ and $0d1$ in the sense of Lemma 5; and that $0e1(0b1)^\omega$ is a word strictly between $0c1$ and $0d1$. Thus by Lemma 5 we have $(0a1)^\omega = 0pu$ and $0e1(0b1)^\omega = 0pv$ where $p := d01c$ and $u, v$ are the appropriate suffixes. Therefore the definition of $\lambda(w, x)$ gives

$$
\begin{aligned}
|\Delta_3| &= \left| \lambda((0a1)^\omega, \bar{b}) - \lambda(0d1(0b1)^\omega, \bar{b}) \right| \\
&= \frac{1}{1-\beta} \left| \begin{array}{l} S(01p, \bar{b}) + \beta^{|01p|} S(u, \phi_{01p}(\bar{b})) - S(10p, \bar{b}) - \beta^{|10p|} S(u, \phi_{10p}(\bar{b})) \\ -S(01p, \bar{b}) - \beta^{|01p|} S(v, \phi_{01p}(\bar{b})) + S(10p, \bar{b}) + \beta^{|01p|} S(v, \phi_{10p}(\bar{b})) \end{array} \right| \\
&= \frac{\beta^{|01p|}}{1-\beta} \left| S(u, \phi_{01p}(\bar{b})) - S(u, \phi_{10p}(\bar{b})) - S(v, \phi_{01p}(\bar{b})) + S(v, \phi_{10p}(\bar{b})) \right| \\
&\leq \frac{\beta^{|01p|}}{1-\beta} \left( \left| S(u, \phi_{01p}(\bar{b})) - S(u, \phi_{10p}(\bar{b})) \right| + \left| S(v, \phi_{01p}(\bar{b})) - S(v, \phi_{10p}(\bar{b})) \right| \right) \\
&\leq \frac{\beta^{|01p|}}{1-\beta} \left( \frac{\underline{a}+1}{1-\beta} + \frac{\bar{b}+1}{1-\beta} \right) \\
&\leq \frac{\beta^{k+1}}{(1-\beta)^2} 2(x+1) \\
&< \frac{\epsilon}{2} \qquad\qquad\qquad\qquad\qquad\qquad\qquad\qquad\qquad\qquad (10)
\end{aligned}
$$

where the last four inequalities follow from the triangle inequality, from Lemma 6, from equation 4 coupled with the fact that $\underline{a} \leq \bar{b} \leq x$ and finally from the definition of $k$.

Finally, coupling 8, 9 and 10 and using the triangle inequality gives

$$
\Delta < \frac{\epsilon}{2} + 0 + \frac{\epsilon}{2} = \epsilon.
$$

This completes the proof. $\qquad\qquad\qquad\qquad\qquad\qquad\qquad\qquad\qquad\qquad\qquad\qquad$ $\square$

## 4 Properties of the Linear-System Orbits $M(w)$

Recall the definitions about words from the main paper, particularly that $\tilde{w}$ is the reverse of $w$. Also, recall the definitions of matrices $F, G, K, M(w)$. The first of the following propositions is used to prove the second. The second appears in the main paper.

**Proposition 11.** *Suppose $w, w'$ are any words. Then*

1. $\det(M(w)) = 1$,

2. $M(\tilde{w}) = K M(w)^{-1} K$,

3. $M(w) = \begin{pmatrix} e & f \\ \frac{eh-1}{f} & h \end{pmatrix}$ *for some $e, f, h \in \mathbb{R}$,*

4. $M(w) - M(\tilde{w}) = \lambda K$ *for some $\lambda \in \mathbb{R}$,*

5. $\dfrac{[M(w01w')]_{22}}{[M(w01w')]_{21}} \geq \dfrac{[M(w10w')]_{22}}{[M(w10w')]_{21}}$,

6. $[M(w)]_{22} \geq [M(w)]_{21}$.

*Proof.* $\det(M(w)) = \prod_{i=1}^{|w|} \det(M(w_i)) = 1$ as $\det(F) = \det(G) = 1$ gives *Claim 1.*

*Claim 2.* The definitions of $F, G, K$ give $KF = F^{-1}K, KG = G^{-1}K$. Thus $KM(w) = M(w_{|w|})^{-1} \cdots M(w_1)^{-1} K = M(\tilde{w})^{-1} K$. The result follows as $K^2 = I$.

*Claim 3.* Put $M(w) =: \begin{pmatrix} e & f \\ g & h \end{pmatrix}$ and solve $\det(M(w)) = 1 = eg - hf$ for $g$.

*Claim 4.* Substituting Claim 2 and Claim 3 in Claim 4 gives $M(w) - KM(w)^{-1}K = (h - e - g)K$.

*Claim 5.* Put $M := M(w), N := M(w')$. We calculate

$$[NGFM]_{22}[NFGM]_{21} - [NGFM]_{21}[NFGM]_{22}$$
$$= (b - a)(M_{11}M_{22} - M_{12}M_{21})((ab + b + a)N_{22}^2 + (b + a + 2)N_{21}N_{22} + N_{21}^2) \geq 0$$

as $b > a \geq 0$, $\det(M) = 1$ and $N \geq 0$. The result follows as $NFGM \geq 0$ and $NGFM \geq 0$.

*Claim 6.* If $w = \epsilon$ then $[M(w)]_{22} - [M(w)]_{21} = 1 \geq 0$. Otherwise we use induction on $|w|$ to show that $M(w)v \geq 0$ where $v := (-1, 1)^T$. In the base case $w \in \{0, 1\}$ so

$$M(w)v = \begin{pmatrix} 1 & 1 \\ c & 1 + c \end{pmatrix} \begin{pmatrix} -1 \\ 1 \end{pmatrix} = \begin{pmatrix} 0 \\ 1 \end{pmatrix} \geq 0 \qquad \text{for some } c \in \{a, b\}.$$

For the inductive step, assume $w = \{0u, 1u\}$ for some word $u$ satisfying $M(u)v \geq 0$. Then

$$M(w)v = \begin{pmatrix} 1 & 1 \\ c & 1 + c \end{pmatrix} M(u)v \geq 0 \qquad \text{for some } c \in \{a, b\}.$$

As $[M(w)v]_2 = [M(w)]_{22} - [M(w)]_{21}$, this completes the proof. $\square$

**Proposition 12.** *Suppose $w$ is a word, $p$ is a palindrome and $n \geq \mathbb{Z}_+$. Then*

1. $M(p) = \begin{pmatrix} \frac{fh+1}{h+f} & f \\ \frac{h^2-1}{h+f} & h \end{pmatrix}$ *for some $f, h \in \mathbb{R}$,*

2. $tr(M(10p)) = tr(M(01p))$,

3. *If $u \in \{p(10p)^n, (10p)^n 10\}$ then $M(u) - M(\tilde{u}) = \lambda K$ for some $\lambda \in \mathbb{R}_-$,*

4. *If $w$ is a prefix of $p$ then $[M(p(10p)^n 10w)]_{22} \leq [M(p(01p)^n 01w)]_{22}$,*

5. $[M((10p)^n 10w)]_{21} \geq [M((01p)^n 01w)]_{21}$,

6. $[M((10p)^n 1)]_{21} \geq [M((01p)^n 0)]_{21}$.

*Proof.* In this proof, we refer to Claim $k$ of Proposition 11 as P$k$.

*Claim 1.* P2 gives $M(p) = KM(p)^{-1}K$ as $p = \tilde{p}$. But in the notation of P3, $[M(p)]_{11} = [KM(p)^{-1}K]_{11}$ says $e = h - (eh - 1)/f$. Solve this for $e$ and substitute in P3.

*Claim 2.* Noting that $GF - FG = (b - a)K$, the notation of Claim 1 gives

$$\text{tr}(M(01p)) - \text{tr}(M(10p)) = \text{tr}(M(p)(GF - GF)) = (b - a)\text{tr}\left(\begin{pmatrix} \frac{fh+1}{h+f} & f \\ \frac{h^2-1}{h+f} & h \end{pmatrix} K\right) = 0.$$

*Claim 3.* Note we can move from $u$ to $\tilde{u}$ just by swapping some 10 for 01. So, repeated application of P5 gives the inequality $\frac{[M(u)]_{22}}{[M(u)]_{21}} \leq \frac{[M(\tilde{u})]_{22}}{[M(\tilde{u})]_{21}}$. But the denominators of this inequality are equal (and non-negative) as P4 gives $[M(u)]_{21} - [M(\tilde{u})]_{21} = \lambda' K_{21} = 0$ for some $\lambda' \in \mathbb{R}$. Thus this inequality reduces to $[M(u)]_{22} \leq [M(\tilde{u})]_{22}$. Yet P4 also gives $[M(u) - M(\tilde{u})]_{22} = \lambda K_{22}$ which combined with the previous sentence says that $\lambda K_{22} \leq 0$. As $K_{22} = 1$, this gives $\lambda \in \mathbb{R}_-$.

*Claim 4.* Let $s$ be the corresponding suffix so $p = ws$ and

$$M(p(10p)^n 10w) - M(p(01p)^n 01w) = M(s)^{-1}(M(p(10p)^{n+1}) - M(p(01p)^{n+1})) =: A.$$

But Claim 3 with $u = p(10p)^{n+1}$ gives

$$[A]_{22} = \underbrace{\lambda[M(s)^{-1}K]_{22}}_{\text{for some } \lambda \leq 0} = \underbrace{[KM(\tilde{s})]_{22}}_{\text{by P2}} = \lambda([M(\tilde{s})]_{22} - [M(\tilde{s})]_{21}) \leq \underbrace{0}_{\text{by P6}}.$$

*Claim 5.* As $M(w) \geq 0$, Claim 3 with $u = (10p)^n 10$ gives

$$[M(w)(M((10p)^n 10) - M((01p)^n 01))]_{21} = \lambda[M(w)K]_{21} = \lambda[-M(w)]_{21} \geq 0.$$

*Claim 6.* Let $E := \begin{pmatrix} 0 & 0 \\ 1 & 1 \end{pmatrix}$. Then $G - F = (b-a)E \geq 0$, so that

$$[GM((10p)^n) - FM((01p)^n)]_{21} = [(b-a)EM((10p)^n) + FM((10p)^n) - FM((01p)^n)]_{21}$$
$$\geq [M((10p)^n 0) - M((01p)^n 0)]_{21} \geq 0$$

by Claim 5. This completes the proof. $\qquad\square$

## 5  Majorisation

In the main paper, we used one result about majorisation which was similar-but-not-identical to any results in Marshall, Olson and Arnold (2011). Let us prove that result.

**Proposition 13.** *Suppose $x, y \in \mathbb{R}_+^m$ and $f : \mathbb{R} \to \mathbb{R}$ is a symmetric function that is convex and decreasing on $\mathbb{R}_+$. Then $x \prec^w y$ and $\beta \in [0,1] \quad \Rightarrow \quad \sum_{i=1}^m \beta^i f(x_{(i)}) \geq \sum_{i=1}^m \beta^i f(y_{(i)})$.*

*Proof.* As the claim relates to $x_{(i)}$ and $y_{(i)}$ we assume that $x_i$ and $y_i$ are in ascending order.

Marshall *et al* (3H2B, page 133) says that if $g : \mathcal{A} \to \mathbb{R}$ is a non-decreasing and convex function on $\mathcal{A} \subseteq \mathbb{R}$ and $(u_1, \ldots, u_m)$ is a non-increasing and non-negative sequence, then for all non-increasing sequences $(p_1, \ldots, p_m)$ the function $\phi(a) := \sum_{i=1}^m u_i g(p_i)$ is Schur-convex.

Indeed the function $f$ is increasing and convex for $p \in \mathbb{R}_-$ (such as $p = -x$ and $p = -y$) and $(\beta, \ldots, \beta^m)$ is a non-increasing and non-negative sequence for $\beta \in [0,1]$. Thus for all non-increasing sequences $(p_1, \ldots, p_m)$ on $\mathbb{R}_-^m$ the function $\psi(p) := \sum_{i=1}^m \beta^i f(p_i)$ is Schur-convex.

Recall (*ibid*, page 12) that $a \in \mathbb{R}^m$ is said to be *weakly submajorised* by $b \in \mathbb{R}^m$, written $a \prec_w b$ if

$$\sum_{i=1}^k a_{[i]} \leq \sum_{i=1}^k b_{[i]}, \quad k = 1, \ldots, m \qquad \text{where } a_{[i]} \text{ denotes } a \text{ in descending order}$$

and that $x \prec_w y \Leftrightarrow -a \prec^w -b$ (*ibid*, page 13).

However (*ibid*, 3A8, page 87) if $\phi(p)$ is a real function on $\mathcal{A} \subset \mathbb{R}^m$ which is non-decreasing in each argument $p_i$ and Schur-convex on $\mathcal{A}$ and $p \prec_w q$ on $\mathcal{A}$ then $\phi(p) \leq \phi(q)$.

Indeed, the function $\psi(p) = \sum_{i=1}^m \beta^i f(p_i)$ is a real function on $\mathbb{R}_-^m$ which is non-decreasing in each argument and Schur-convex on $\mathbb{R}_-^m$ for all non-increasing sequences $(p_1, \ldots, p_m)$. Furthermore, $-y \prec_w -x$ as $x \prec^w y$. Therefore $\psi(y) = \psi(-y) \leq \psi(-x) = \psi(x)$ as claimed. $\qquad\square$

## 6  Clarification of Theorem 1 for $0 \leq x \leq y_1$ or $y_0 \leq x < \infty$

Recall the following definitions and assumption from the main paper

$$F := \begin{pmatrix} 1 & 1 \\ a & 1+a \end{pmatrix}, \quad G := \begin{pmatrix} 1 & 1 \\ b & 1+b \end{pmatrix}, \quad E := \begin{pmatrix} 0 & 0 \\ 1 & 1 \end{pmatrix}, \quad v(x) := \begin{pmatrix} z \\ 1 \end{pmatrix}, \quad b > a \geq 0.$$

If $0 \leq x \leq y_1$ or $y_0 \leq x < \infty$ then the relevant linear systems, (9) in the main paper, are

$$\left. \begin{array}{l} (M(1^{k+1}) - M(01^k))v(x) = (G-F)G^k v(x) = (b-a)EG^k v(x) \geq 0 \\ (M(10^k) - M(0^{k+1}))v(x) = (G-F)F^k v(x) = (b-a)EF^k v(x) \geq 0 \end{array} \right\} \quad \text{for } k \in Z_+$$

where both inequalities follow as $E, F, G$ are all $\geq 0$, as $b > a$ and as $x \geq \min\{y_0, y_1\} \geq 0$. Therefore all cumulative sums of the above expressions are non-negative so the derivative of the numerator of the Whittle index is non-negative by the same weak-supermajorisation argument as in the main paper.

Meanwhile, the denominator of the index in these cases is

$$\sum_{k=0}^\infty \beta^k((1^\omega)_{k+1} - (01^\omega)_{k+1}) = \beta = \sum_{k=0}^\infty \beta^k((10^\omega)_{k+1} - (0^\omega)_{k+1})$$

which is non-negative. Therefore the rest of the proof of Theorem 1 follows as in the main paper.

In fact we could say that the majorisation point, which is $\phi_w(0)$ for words $0w1$ in the main paper, is $-1$ in both cases. Indeed, Claim 6 of Proposition 4 of the main paper says that $Fv(-1) = Gv(-1) = v(0)$. Also, $Ev(-1) = (0, 0)^T$. Thus for all $k \in \mathbb{Z}_+$, $EG^k v(-1) \geq EF^k v(-1) \geq 0$ whereas $Ev(-1-\epsilon) < 0$ for any $\epsilon > 0$.

# 7 Computational Cost

In this section we consider the complexity of approximating the index for real-number and arbitrary-precision models of computation.

## 7.1 Real-Number Model of Computation

In a real-number model of computation, all rational functions with real-valued coefficients of a given real-valued input can be computed exactly in a time proportional to the number of basic arithmetical operations $(+, -, \times, /)$ and simple functions (such as $\log(\cdot)$ and $\lfloor \cdot \rfloor$) involved. It is straightforward to approximate the Whittle index given by Equations (5) and (6) of the main paper to any given tolerance $\epsilon > 0$ by truncating the sums in the numerator and denominator to a appropriate number of terms $T$. When we say "to a given tolerance" we mean that $\left| \lambda^W(x) - \hat{\lambda}(x) \right| \leq \epsilon$ where $\lambda^W(\cdot)$ is given by Equation (6) and $\hat{\lambda}(\cdot)$ is our approximation.

To make such an approximation, we first define a function which gives the optimal state and action sequence of a given point under mapping (5).

---

**Algorithm:** $[x_{0:T}, u_{0:T}] \leftarrow \textbf{orbit}(x_0, u_0, \phi_0, \phi_1, T)$
**Input:** Initial point $x_0 \in \mathbb{R}_+$ and action $u_0 \in \{0, 1\}$, mappings $\phi_0, \phi_1 : \mathbb{R}_+ \to \mathbb{R}_+$,
number of terms $T \in \mathbb{Z}_+$.
**Output:** State sequence $x_{0:T}$ and action sequence $u_{0:T}$ for Equation (5)

1: **for** $t = 1, \ldots, T$
2: $\quad x_t \leftarrow \phi_{u_{t-1}}(x_{t-1})$
3: $\quad u_t \leftarrow \begin{cases} \mathbf{1}_{x_t > x_0} & \text{if } u_0 = 0 \\ \mathbf{1}_{x_t \geq x_0} & \text{otherwise} \end{cases}$
4: **end**

---

Since the index itself is a ratio of two sums, it is helpful to overestimate the denominator to ensure a good approximation guarantee. The following algorithm approximates the index, ensuring such overestimation by including a term $\beta^{T+1}$ in the denominator $D_T$ in certain cases.

---

**Algorithm:** $\hat{\lambda} \leftarrow \textbf{ApproximateWhittleIndex}(x, \beta, \phi_0, \phi_1, \epsilon)$
**Input:** Initial point $x \in \mathbb{R}_+$, discount $\beta \in (0, 1)$, mappings $\phi_0, \phi_1 : \mathbb{R}_+ \to \mathbb{R}_+$,
tolerance $\epsilon \in \mathbb{R}_{++}$
**Output:** Approximation $\hat{\lambda}$ to the Whittle index in Equation (6)
assuming weight $w = 1$ and observation cost $h = 0$

1: $T \leftarrow \max\left\{ 0, \left\lfloor \left( \log \frac{2(x+1)}{\epsilon(1-\beta)^3} \right) / \left( \log \frac{1}{\beta} \right) \right\rfloor \right\}$
2: $\begin{cases} [x_{0:T}^{0*}, u_{0:T}^{0*}] \leftarrow \textbf{orbit}(x, 0, T) \\ [x_{0:T}^{1*}, u_{0:T}^{1*}] \leftarrow \textbf{orbit}(x, 1, T) \end{cases}$
3: $N_T \leftarrow \sum_{t=0}^{T} \beta^t (x_t^{0*} - x_t^{1*})$
4: $D_T \leftarrow \mathbf{1}_{u_T^{1*} - u_T^{0*} \neq 1} \beta^{T+1} + \sum_{t=0}^{T} \beta^t (u_t^{1*} - u_t^{0*})$
5: $\hat{\lambda} \leftarrow N_T / D_T$

---

We are now ready to state the corresponding approximation guarantee.

**Proposition 14.** ApproximateWhittleIndex *approximates the Whittle index for Problem KF1 to within $\epsilon > 0$ in* $O\left(\left(\log\frac{2(x+1)}{\epsilon(1-\beta)^3}\right) \Big/ \left(\log\frac{1}{\beta}\right)\right)$ *operations.*

*Proof.* The computational cost is immediate, since each call to orbit involves $O(T)$ operations as does each of the sums giving $N_T, D_T$. Thus the total cost is $O(T)$ where $T$ is given in line 1 of ApproximateWhittleIndex.

It remains to show that the approximation is to a tolerance of $\epsilon$. Recall the full numerator and denominator of $\lambda^W(\cdot)$

$$N := \sum_{t=0}^{\infty} \beta^t (x_t^{0*} - x_t^{1*}), \qquad\qquad D := \sum_{t=0}^{\infty} \beta^t (u_t^{1*} - u_t^{0*}).$$

First, we bound $N$ and the error $|N - N_T|$. Each term $x_t^{0*}$ and $x_t^{1*}$ is in $[0, \phi_0(x)]$ and $\phi_0(x) \le x+1$ so

$$N \le \sum_{t=0}^{\infty} \beta^t \left|x_t^{0*} - x_t^{1*}\right| \le \sum_{t=0}^{\infty} \beta^t (x+1) = \frac{x+1}{1-\beta}.$$

Similarly

$$|N_T - N| \le \sum_{t=T+1}^{\infty} \beta^t (x+1) = \beta^{T+1}\frac{x+1}{1-\beta}.$$

Second, we bound $D$ and the error $|D - D_T|$. As $u_t^{1*}$ and $u_t^{0*}$ form the sequences $(10w)^\omega$ and $(01w)^\omega$ respectively for some mechanical word $0w1$, the sequence $(\Delta u_t := u_t^{1*} - u_t^{0*} : t = 0, 1, \dots)$ is $(1, -1, 0^{|w|})^\omega$. Therefore

$$D = \sum_{k=0}^{\infty} \beta^{k|0w1|}(1-\beta) = \frac{1-\beta}{1-\beta^{|0w1|}} \ge 1-\beta.$$

For the same reason, if $\Delta u_T = 1$ then $(\Delta u_{T+1}, \Delta u_{T+2}, \dots) = (-1, 0^{|w|}, 1, -1, \dots)$, so

$$D_T - D = \beta^{T+1} - \beta^{T+2} \sum_{t=0}^{\infty} \beta^t \Delta u_{T+2+t} = \beta^{T+1}\left(1 - \beta^n \frac{1-\beta}{1-\beta^{|0w1|}}\right)$$

for some $n \ge 0$. Also, if $\Delta u_T \ne 1$ then ApproximateWhittleIndex includes an extra term giving

$$D_T - D = \beta^{T+1} - \beta^{T+1} \sum_{t=0}^{\infty} \beta^t \Delta u_{T+1+t} = \beta^{T+1}\left(1 - \beta^{n'} \frac{1-\beta}{1-\beta^{|0w1|}}\right)$$

for some $n' \ge 0$. But for $n'' \in \{n, n'\}$ we have

$$0 \le \beta^{n''}\frac{1-\beta}{1-\beta^{|0w1|}} \le \beta^{n''}\frac{1-\beta}{1-\beta^2} = \frac{\beta^{n''}}{1+\beta} \le 1$$

so whatever the value of $\Delta u_T$ we conclude that

$$D_T - D \in [0, \beta^{T+1}].$$

Finally, coupling the above bounds gives

$$\left| \frac{N}{D} - \frac{N_T}{D_T} \right| = \left| \frac{N(D_T - D) - (N_T - N)D}{DD_T} \right|$$

$$\leq \left| \frac{\frac{N}{D}(D_T - D) - (N_T - N)}{D} \right| \tag{11}$$

$$\leq \frac{\frac{N}{D}|D_T - D| + |N_T - N|}{D} \tag{12}$$

$$\leq \frac{\frac{x+1}{1-\beta}\beta^{T+1} + \beta^{T+1}\frac{x+1}{1-\beta}}{1-\beta} \tag{13}$$

$$= \beta^{T+1} \frac{\frac{1}{(1-\beta)^2} + \frac{1}{1-\beta}}{1-\beta}(x+1)$$

$$\leq \beta^{T+1} \times \frac{2(x+1)}{(1-\beta)^3}$$

$$\leq \exp\left\{ \left( -\log\frac{1}{\beta} \right) \left( \log\frac{2(x+1)}{\epsilon(1-\beta)^3} \right) / \left( \log\frac{1}{\beta} \right) \right\} \times \frac{2(x+1)}{(1-\beta)^3} \tag{14}$$

$$= \exp\left\{ \log\frac{\epsilon(1-\beta)^3}{2(x+1)} \right\} \times \frac{2(x+1)}{(1-\beta)^3}$$

$$= \epsilon$$

where 11 follows from $D_T \geq D \geq 0$, 12 follows from the triangle inequality and the fact that $N \geq 0$ and $D \geq 0$, 13 follows from the bounds derived just above and 14 follows from the definition of $T$ in Algorithm ApproximateWhittleIndex. This completes the proof. □

## 7.2 Arbitrary-Precision Computation

In an arbitrary-precision model of computation, functions of arbitrary-precision numbers may be computed to tolerance $\epsilon$ in a number of operations proportional to the number of basic arithmetic operations involved $(+, -, \times, /)$ times the work required for one multiplication $M(\epsilon)$ which itself varies with the tolerance. When approximating the Whittle index in such a model, we must face the complication that iterates $x_t$ (as in the function orbit above) may be so close to the threshold $x$ that an arbitrarily small tolerance is required to correctly decide whether $x_t > x$. We avoid this arbitrarily large computational cost by allowing some wrong decisions and compensating for such wrong decisions via the continuity of the index (Proposition todo:continuity) and a bound on the gradient of approxmations to the index based on partial sums.

First we define some notation and present a suitable algorithm, APIndex. Then we sketch an analysis of the quality of approximation and of the computational cost. These sketches simply aim to demonstrate that there is an efficient algorithm for the problem. We believe that there are more efficient but more complicated algorithms for this problem.

**Notation.** We remind the reader that $\tilde{w}$ is the reverse of word $w$ and $w^\omega = www\cdots$ is the result of infinitely concatenating $w$. Also, recall that $y_v$ is the fixed point of word $v$ given by the solution to $\phi_v(y) = y$ with $y \in \mathbb{R}_+$.

Let $\mathcal{W}$ denote the set of mechanical words and $\mathcal{W}_{\leq T} \subset \mathcal{W}$ denote the set of mechanical words of length at most $T \in \mathbb{Z}_{++}$. For given $\phi_0, \phi_1$ and $T \in \mathbb{Z}_{++}$ let $W_T : \mathbb{R}_+ \to \mathcal{W}$ be the function that maps point $x \in \mathbb{R}_+$ to the word $w_1 \ldots w_T$ where

$$x_1 := \phi_1(x), \qquad w_1 := \mathbf{1}_{x_1 \geq x}, \qquad x_t := \phi_{w_{t-1}}(x_{t-1}), \qquad w_t := \mathbf{1}_{x_t \geq x}$$

for $t = 2, 3, \ldots, T$, which is of course the mapping in the function orbit above.

**Algorithm.** There are two keys to our algorithm.

First, we use a function $f : \mathbb{R}_+ \times \mathscr{W} \times \mathbb{Z}_+ \to \mathbb{R}_+$ defined as

$$f(x, w, T) := \frac{1 - \beta^{|w|}}{1 - \beta} \sum_{t=0}^{T} \beta^t (\phi_{(0\tilde{w}^\omega)_{1:t}}(x) - \phi_{(1w^\omega)_{1:t}}(x)). \qquad (15)$$

If we recall that the denominator of the index is $\frac{1-\beta}{1-\beta^{|w|}}$ (see the proof of Theorem 1), it is easy to see that $\lim_{T\to\infty} f(x, W_T(x), T)$ equals the Whittle index $\lambda^W(x)$.

Second, we use the following explicit representation of $W_T(\cdot)$.

**Lemma 10.** *Suppose $w^{(1)}, w^{(2)}, \ldots, w^{(n)}$ are the words of $\mathscr{W}_{\leq T}$ in order of decreasing rate $|w|_1 / |w|$ and let $x^{(k)}$ be the upper fixed point of word $w^{(k)}$ for $k = 1, 2, \ldots, n - 1$. Let $p(w) := (w^\omega)_{1:T}$ be the length-$T$ prefix generated by repeating word $w$. Then*

$$W_T(x) = \begin{cases} p(w^{(1)}) & \text{for } x \in [0, x^{(1)}] \\ p(w^{(k)}) & \text{for } x \in (x^{(k-1)}, x^{(k)}] \text{ and } k = 2, 3, \ldots, n - 1 \\ p(w^{(n)}) & \text{for } x \in (x^{(n-1)}, \infty). \end{cases}$$

*Example.* If $T = 4$ then the elements of $\mathscr{W}_{\leq T}$ are

$$(w^{(1)}, w^{(2)}, \ldots, w^{(n)}) = (1, 0111, 011, 01, 001, 0001, 0)$$

which generate the following length-$T$ prefixes

$$(p(w^{(1)}), p(w^{(2)}), \ldots, p(w^{(n)})) = (1111, 0111, 0110, 0101, 0010, 0001, 0000)$$

and Proposition 2 of the main paper shows the upper fixed points are

$$(x^{(1)}, x^{(2)}, \ldots, x^{(n-1)}) = (y_1, y_{1011}, y_{101}, y_{10}, y_{100}, y_{1000}).$$

*Proof. (Sketch.)* The result follows from Proposition 2 once we demonstrate that the prefixes generated by $\mathscr{W}$ are the same as those generated by $\mathscr{W}_{\leq T}$, so that

$$\{p(w) : w \in \mathscr{W}\} = \{p(w) : w \in \mathscr{W}_{\leq T}\}.$$

One way of seeing this is to show that any length-$k$ prefix generated by a word $uv$ corresponding to a node $(u, v)$ of the Christoffel tree is also generated by an ancestor of that node, provided $k < |uv|$. Specifically, proper prefixes from $(uv, v)$ are also generated by $(u, v)$ and proper prefixes of $(u, uv)$ are also generated by whichever ancestor generated the proper prefixes of $(u, v)$. $\qquad\square$

In the pseudocode that follows, we first choose a large enough number of terms $T$ for the partial sum defining $f(\cdot, \cdot, \cdot)$. Unfortunately, as remarked above, we may not be able to compute $W_T(x)$ efficiently. Nevertheless it is possible to efficiently compute $W_T(z)$ for some $z$ that is close enough to $x$ that $|f(x, W_T(z), T) - f(x, W_T(x), T)|$ and hence $|f(x, W_T(z), T) - \lambda^W(x)|$ are small. So the second step of the algorithm is to select a maximum error $\delta$ such that any word $W_T(z)$ with $|x - z| \leq \delta$ will give a good-enough approximation. Then we loop through the set $\mathscr{W}_{\leq T}$ to essentially select such a word $W_T(z)$ to which we apply function $f(\cdot, \cdot, \cdot)$. We say "essentially" because we actually select a word $w$ in $\mathscr{W}_{\leq T}$ but it follows from the explicit representation of $W_T(z)$ given above that $W_T(z) = p(w)$ and from the definition of $f(\cdot, \cdot, \cdot)$ that $f(\cdot, p(w), T) = f(\cdot, w, T)$ for any word $w$.

**Algorithm:** $\hat{\lambda} \leftarrow \textbf{APIndex}(x, \phi_0, \phi_1, \beta, \epsilon)$
**Input:** $x \in \mathbb{R}_+$, mappings $\phi_0, \phi_1 : \mathbb{R}_+ \rightarrow \mathbb{R}_+$, discount $\beta \in (0, 1)$, tolerance $\epsilon \in \mathbb{R}_+$
**Output:** $\hat{\lambda} \in \mathbb{R}_+$ with $|\hat{\lambda} - \lambda^W(x)| \leq \epsilon$ where $\lambda^W(\cdot)$ is as in Equation (6)

1: $T \leftarrow$ a positive integer with $\beta^T \times 2(x+1)/(1-\beta)^2 \leq \epsilon/(3T^2)$
2: $\delta \leftarrow$ a positive real number with $g\delta \leq \epsilon/6$ where $g := \beta/(1-\beta)^2$
3: **for** each word $w^{(k)} \in \mathscr{W}_{\leq T} \backslash \{0\}$
4: $\quad x^{(k)} \leftarrow$ the upper fixed point of $w^{(k)}$ to tolerance $\delta/2$
5: **end**
6: $w \leftarrow$ the word $w^{(k)}$ in $\mathscr{W}_{\leq T}$ with largest rate $|w|_1 / |w|$, satisfying $x \leq x^{(k)}$ to tolerance $\delta/2$
7: $\hat{\lambda} \leftarrow f(x, w, T)$ to tolerance $\epsilon/3$ where $f(\cdot, \cdot, \cdot)$ is given by 15

**Quality of Approximation.** We show that APIndex approximates the Whittle index to tolerance $\epsilon$ using the triangle inequality, relying on the following four facts.

*Fact 1.* For any fixed mechanical word $w$, the gradient of $f(z, w, T)$ for any $z \in$ is bounded by

$$\left| \frac{df(z, w, T)}{dz} \right| \leq \frac{\sum_{t=0}^{T} \beta^t \left| \frac{d}{dz} \phi_{(0\tilde{w}^\omega)_{1:t}}(z) - \frac{d}{dz} \phi_{(1w^\omega)_{1:t}}(z) \right|}{\frac{1-\beta}{1-\beta^{|w|}}} \leq \frac{\sum_{t=1}^{T} \beta^t}{1-\beta} \leq \frac{\beta}{(1-\beta)^2} =: g.$$

*Fact 2.* If $T$ is chosen such that $f(x, W_T(x), T)$ is an $\epsilon/(3T^2)$ approximation to $\lambda^W(x)$ then the value of $f(\cdot, \cdot, \cdot)$ at the upper fixed points satisfies

$$\left| f(x^{(k-1)}, w^{(k-1)}, T) - f(x^{(k-1)}, w^{(k)}, T) \right| \leq \left| f(x^{(k-1)}, w^{(k-1)}, T) - \lambda^W(x^{(k-1)}) \right|$$
$$+ \left| \lambda^W((x^{(k-1)})^+) - f(x^{(k-1)}, w^{(k)}, T) \right| \leq \frac{2\epsilon}{3T^2}$$

where $(x^{(k-1)})^+$ denotes the point just larger than $x^{(k-1)}$. This follows from the triangle inequality and because $\lambda^W(\cdot)$ is a continuous function.

*Fact 3.* The number of words in $\mathscr{W}_{\leq T}$ satisfies

$$|\mathscr{W}_{\leq T}| = 1 + \sum_{i=1}^{T} \varphi(i) \leq 1 + \sum_{i=1}^{T} (i-1) \leq \frac{T^2}{2} \quad \text{for } T \geq 2$$

for the following reasons. The equality follows from the fact that the number of mechanical words of length $T$ is 2 for $T = 1$ (the words 0,1) and equals Euler's totient function $\varphi(T)$ for $T > 1$. The latter fact is readily seen from the definition of a Christoffel $0w1$ in terms of any pair of relatively prime integers $|0w1|_0$ and $|0w1|_1$. The inequality follows from the fact that $\varphi(i)$ is largest when $i$ is prime in which case it equals $i - 1$.

*Fact 4.* If $T$ is chosen as in Line 1 of the algorithm, then $f(x, W_T(x), T)$ approximates $\lambda^W(x)$ to tolerance $\epsilon/(3T^2)$. This follows from the same analysis as presented for algorithm Approximate-WhittleIndex.

Now for the triangle inequality. Suppose $x \in (x^{(j-1)}, x^{(j)}]$ but APIndex sets $w \leftarrow w^{(i)}$ on Line 6. Suppose also that $x^{(j)} < x^{(i)}$, but note that a symmetric result holds for $x^{(j)} > x^{(i)}$. Then

$$
\left| \lambda^W(x) - \hat\lambda \right|
$$

$$
\leq \underbrace{\left| \lambda^W(x) - f(x, w^{(j)}, T) \right|}_{\leq\, \epsilon/(3T^2) \text{ by Fact 4}} + \underbrace{\left| f(x, w^{(j)}, T) - f(x^{(j)}, w^{(j)}, T) \right|}_{\leq\, g\left|x^{(j)} - x\right| \text{ by Fact 1}}
$$

$$
+ \underbrace{\left| f(x^{(j)}, w^{(j)}, T) - f(x^{(j)}, w^{(j+1)}, T) \right|}_{\leq\, 2\epsilon/(3T^2) \text{ by Fact 2}} + \underbrace{\left| f(x^{(j)}, w^{(j+1)}, T) - f(x^{(j+1)}, w^{(j+2)}, T) \right|}_{\leq\, g\left|x^{(j+1)} - x^{(j)}\right| \text{ by Fact 1}} + \ldots
$$

$$
+ \underbrace{\left| f(x^{(i-1)}, w^{(i-1)}, T) - f(x^{(i-1)}, w^{(i)}, T) \right|}_{\leq\, 2\epsilon/(3T^2) \text{ by Fact 2}} + \underbrace{\left| f(x^{(i-1)}, w^{(i)}, T) - f(x, w^{(i)}, T) \right|}_{\leq\, g\left|x^{(i-1)} - x\right| \text{ by Fact 1}}
$$

$$
+ \underbrace{\left| f(x, w^{(i)}, T) - \hat\lambda \right|}_{\leq\, \epsilon/3 \text{ by Line 7}}
$$

$$
\leq \underbrace{\left(1 + 2(j-i)\right)}_{\leq\, T^2 \text{ by Fact 3}} \epsilon/(3T^2) + 2g \underbrace{\left|x^{(i-1)} - x\right|}_{\leq\, \delta \text{ by Lines 4 and 6}} + \epsilon/3
$$

$$
\leq \frac{\epsilon}{3} + \frac{\epsilon}{3} + \frac{\epsilon}{3} = \epsilon
$$

where in the last inequality we used the definition of $g$ in Fact 1 and the choice of $\delta$ in Line 2.

**Computational Cost.** Suppose the input satisfies $x < 2^n$ and $\epsilon > 2^{-m}$. For brevity we shall consider $\beta \in (0,1)$ to be fixed and ignore factors in $\beta$. Further, we suppose that arithmetic operations on positive numbers less than $2^n$ to tolerance $2^{-m}$ require $M(n+m)$ operations for an appropriate function $M(\cdot)$.

The analysis is as follows. On Line 1 setting $T = O(\log(x/\epsilon)) = O(n+m)$ suffices and on Line 2 setting $\delta = \Omega(\epsilon)$ suffices. Now, for Lines 3-5, it follows from Fact 3 that we have to approximate $O(T^2)$ fixed points. This may be done by traversing the Christoffel tree and exploring $O(T^2)$ nodes. At each node, we may compute the fixed point of the corresponding word $0w1$ by solving the quadratic equation $\phi_{10w}(y) = y$. To formulate this equation we may start by finding a matrix corresponding to the Möbius transformation $\phi_{10w}(\cdot)$. Indeed such a matrix can be constructed as a product of $O(T)$ of the matrices $M_0, M_1$ that define $\phi_0(\cdot)$ and $\phi_1(\cdot)$. As we require a tolerance of $\Omega(\delta) = \Omega(\epsilon)$, this product requires $O(TM(n+m))$ operations. Thus solving the quadratic equation requires $O(TM(n+m))$ operations, since approximating the square root of an $n$-bit input requires $O(M(n))$ operations. In summary Lines 3-5 require $O(T^3 M(n+m))$ operations.

On Line 6 we may need $O(T^2)$ comparisons of numbers smaller than $2^n$ to tolerance $\delta = \Omega(\epsilon)$ requiring $O(T^2 M(n+m))$ operations. Finally, Line 7 involves a sum of $T$ terms, each requiring the application of a Möbius transformation involving numbers less than $2^n$ to tolerance $\Omega(\epsilon)$. This requires $O(TM(n+m))$ operations.

In conclusion, Lines 3-5 dominate so the total computational cost is

$$
O(T^3 M(n+m)) = O((n+m)^3 M(n+m)).
$$