[Reviews · NeurIPS 2015]

Submitted by Assigned_Reviewer_1

This paper attempts to answer a question of "indexability" for simple Kalman filter, restless bandits. The context used is that of tracking N independent time-series, each of which follows a Gaussian random walk. At each time point (discrete time) an action is made to observe one of these time-series with higher precision than the others. The task is to choose the actions (or a policy producing the actions) which minimises a weighted sum of the error variances. The authors cast this problem as a multi-arm restless bandit and ask if previous work on "indexing" can be applied to it. The main result of the paper is to prove that the bandit is indexable under certain assumptions on the optimal policy.

The first part of the paper lays out the problem as has just been described, as well as stating the main theorem of the paper. The authors do this with a reasonable amount of clarity. That said, some sentences could be written more clearly (e.g. the last sentence of the opening paragraph).

The bulk of the paper is then made up of a proof of the main theorem. This involves the use of mathematical concepts which I suspect are relatively unknown amongst the community. The authors do a commendable job of trying to introduce all the definitions and lead the reader through the proof.

While the result of the paper is interesting I feel the authors could have done more to explain why it is important. Clearly there is a significant limit on space due to the necessity of introducing the mathematical concepts, but there is nothing in the paper to tie the result back to the original context in which the problem was set.
Summary: This paper establishes a result on "indexability" for a simple class of restless bandits, providing a complex proof of the main theorem. Whilst the paper is mathematically detailed, and the result interesting, more could have been done to explain the significance of the result.

Submitted by Assigned_Reviewer_2

It would be desirable to see more corollaries / consequences from the main result.

The authors should consider to rewrite the paper in a way that a person with some standard background on bandits and Kalman filters can understand the main result and the intuition behind the proofs.
Summary: This paper studies a very simple restless bandit problem where estimates are updated via scalar Kalman filters. To my assessment, this paper is very interesting but difficult to access.

Submitted by Assigned_Reviewer_3

The paper uses majorization over the action tree to show that simple kalman filters are indexable. Along the way, the paper discusses some facts which are not central and not necessarily useful.

Quality: The end result is interesting.

Clarity: The name dropping of so many terms will hurt the readership of this paper. Making a lot of mathematical analogies is not the same as doing mathematics.

Originality: The end result is new to the best of knowledge of the reviewer.

Significance: The result can be significant.
Summary: The paper makes progress on an important problem. However the paper is also distracting because it is often more interested in showing off and making random comments in passing.

Author Feedback
Author rebuttal: Dear Reviewers,

Thank you for your reviews. As some of you hinted, the paper may not be among the easiest to read, even if we worked hard to make it readable within page constraints. This said, we are encouraged by the fact that the highest scores were given by the most confident reviewers.

Part of the possible difficulty with the paper is that it uses concepts that are not commonly used in related proofs. While this may add some burden to the reader, it was also partly the idea of the paper to introduce a new set of tools that could be used to tackle other related problems. We see this as a methodological contribution of the paper in addition to the actual main result about the indexability. Indeed, we address a fairly long-standing problem and if commonly-used tools were sufficient to solve this problem, someone else would probably already have already solved it.

Some of the concerns raised were about further elaborating on the relevance of the results to the NIPS audience. Although the relation of our work to active learning and attention is evident and more generally the optimal decision on what-to-attend-to-next can be found in many contexts like in the cited work on analyzing data streams, we only briefly listed such topics in the introduction. Therefore we shall describe those topics further and to make the matter concrete we shall give a numerical comparison of the performance of the Whittle index policy with heuristic policies in the Supplementary Material.

The issue of complexity was not addressed as we are still working on a satisfactory answer. So far we can say the following.

1. In the Blum-Shub-Smale rational-number-computation model, given rational input, rational comparisons are accurate and have unit cost. So we can be sure to make the right comparisons (right-choice-of-word(!)) in equation (5). Further, the n^th partial sums of the numerator and denominator of the index for input x given the right-choice-of-word(!) are of the form \sum_{t=0}^n \beta^t a_t, where \abs{a_t} is O(x). So, basic arguments lead us conclude that the number of terms required to compute the index to within a given error \epsilon is O(log(1/\epsilon) log(x+1)) for fixed discount \beta<1. However, we wish to make this into a bound on the cost to attain a given relative error and extend it to non-zero observation costs h in (6).

2. In models where real-number multiplication costs M(b) for b-bit input-or-output (e.g. Brent and Zimmermann, 2010), we may-or-may-not require lots of work to make the right-choice-of-word(!). Indeed the continuity proof we presented suggests a cost that is exponential in the bit-length b of the input x, since the interval-lengths l_k (Lemma~9, Supplementary Material) can shrink rapidly with depth in the Christoffel tree (e.g. consider inputs x close to the fixed point of the Fibonacci word).

3. The conclusion suggested by statement-2 may be pessimistic, as our continuity proof was not intended for this purpose. One escape seems to be to consider "the tree generated by all possible length-b prefixes of Sturmian words". Using results in Lothaire, it appears that there are only 1+\sum_{k=1}^b EulerTotient(k) such length-b prefixes and the corresponding interval-lengths shrink more slowly than in statement-2. This suggests polynomial complexity in log(1/epsilon) and b for fixed \beta<1, but our proof thereof is currently incomplete.

So, we will address complexity issues and further elaborate on the diverse applications of this work to multiple problem families addressed by the community in the final version of the paper.